# Ringmaster ASGD: The First Asynchronous SGD with Optimal Time Complexity

Artavazd Maranjyan [1]  Alexander Tyurin [2 3]  Peter Richtárik [1]

## Abstract

Asynchronous Stochastic Gradient Descent (Asynchronous SGD) is a cornerstone method for parallelizing learning in distributed machine learning. However, its performance suffers under arbitrarily heterogeneous computation times across workers, leading to suboptimal time complexity and inefficiency as the number of workers scales. While several Asynchronous SGD variants have been proposed, recent findings by Tyurin & Richtárik (2023) reveal that none achieve optimal time complexity, leaving a significant gap in the literature. In this paper, we propose Ringmaster ASGD, a novel Asynchronous SGD method designed to address these limitations and tame the inherent challenges of Asynchronous SGD. We establish, through rigorous theoretical analysis, that Ringmaster ASGD achieves optimal time complexity under arbitrarily heterogeneous and dynamically fluctuating worker computation times. This makes it the first Asynchronous SGD method to meet the theoretical lower bounds for time complexity in such scenarios.

## 1. Introduction

We consider stochastic nonconvex optimization problems of the form

$$\min_{x \in \mathbb{R}^d} \left\{ f(x) := \mathbb{E}_{\xi \sim \mathcal{D}} \left[ f(x; \xi) \right] \right\},$$

where $f : \mathbb{R}^d \times \mathbb{S}_\xi \to \mathbb{R}$, $\mathbb{R}^d$ is a linear space, and $\mathbb{S}_\xi$ is a sample space. In machine learning, $f(x; \xi)$ denotes the loss of a model parameterized by $x$ on a data sample $\xi$, and $\mathcal{D}$ denotes the distribution of the training dataset. In nonconvex

[1] King Abdullah University of Science and Technology, Thuwal, Saudi Arabia [2] AIRI, Moscow, Russia [3] Skolkovo Institute of Science and Technology, Moscow, Russia. Correspondence to: Artavazd Maranjyan <https://artomaranjyan.github.io/>.

*Proceedings of the 42nd International Conference on Machine Learning*, Vancouver, Canada. PMLR 267, 2025. Copyright 2025 by the author(s).

optimization, our goal is to find an $\varepsilon$–stationary point, i.e., a (random) vector $\bar{x} \in \mathbb{R}^d$ such that $\mathbb{E}[\|\nabla f(\bar{x})\|^2] \leq \varepsilon$.

We consider a setup involving $n$ workers (e.g., CPUs, GPUs, servers), each with access to the same distribution $\mathcal{D}$. Each worker is capable of computing independent, unbiased stochastic gradients with bounded variance (Assumption 1.3). We consider a setup with asynchronous, heterogeneous, and varying computation speeds. We aim to account for all potential scenarios, such as random outages, varying computational performance over time, and the presence of slow or straggling workers (Dean & Barroso, 2013).

This setup is common in both datacenter environments (Dean et al., 2012) and federated learning (Konečný et al., 2016; McMahan et al., 2016; Kairouz et al., 2021) for distributed training. Although parallelism facilitates rapid convergence, variations in worker speeds make effective coordination more challenging.

Asynchronous Stochastic Gradient Descent (Asynchronous SGD) is a popular approach for parallelization in such distributed settings. The algorithm operates as follows (Algorithm 1), formalized below.

---
**Algorithm 1** Asynchronous SGD

**Input:** point $x^0 \in \mathbb{R}^d$, stepsizes $\gamma_k \geq 0$
Set $k = 0$
Workers start computing stochastic gradients at $x^0$
**while** True **do**
    Gradient $\nabla f(x^{k-\delta^k}; \xi_i^{k-\delta^k})$ arrives from worker $i$
    Update: $x^{k+1} = x^k - \gamma_k \nabla f(x^{k-\delta^k}; \xi_i^{k-\delta^k})$
    Worker $i$ begins calculating $\nabla f(x^{k+1}; \xi_i^{k+1})$
    Update the iteration number $k = k + 1$
**end while**

---

This is a greedy and asynchronous method. Once a worker finishes computation of the stochastic gradient, it immediately sends the gradient to the server, which updates the current iterate without waiting for other workers. Notice that, unlike vanilla SGD, the update is performed using the stochastic gradient calculated at the point $x^{k-\delta^k}$, where the index $k - \delta^k$ corresponds to the iteration when the worker started computing the gradient, which can be significantly

outdated. The sequence $\{\delta^k\}$ is a sequence of delays of Asynchronous SGD, where $\delta^k \geq 0$ is defined as the difference between the iteration when worker $i$ started computing the gradient and iteration $k$, when it was applied.

Asynchronous SGD methods have a long history, originating in 1986 (Tsitsiklis et al., 1986) and regaining prominence with the seminal work of Recht et al. (2011). The core idea behind Asynchronous SGD is simple: to achieve fast convergence, all available resources are utilized by keeping all workers busy at all times. This principle has been validated in numerous studies, showing that Asynchronous SGD can outperform naive synchronous SGD methods (Feyzmahdavian et al., 2016; Dutta et al., 2018; Nguyen et al., 2018; Arjevani et al., 2020; Cohen et al., 2021; Mishchenko et al., 2022; Koloskova et al., 2022; Islamov et al., 2024; Feyzmahdavian & Johansson, 2023).

## 1.1. Assumptions

In this paper, we consider the standard assumptions from the nonconvex world.

**Assumption 1.1.** *Function $f$ is differentiable, and its gradient is $L$–Lipschitz continuous, i.e.,*

$$\|\nabla f(x) - \nabla f(y)\| \leq L \|x - y\|, \ \forall x, y \in \mathbb{R}^d.$$

**Assumption 1.2.** *There exist $f^{\inf} \in \mathbb{R}$ such that $f(x) \geq f^{\inf}$ for all $x \in \mathbb{R}^d$. We define $\Delta := f(x^0) - f^{\inf}$, where $x^0$ is the starting point of optimization methods.*

**Assumption 1.3.** *The stochastic gradients $\nabla f(x; \xi)$ are unbiased and have bounded variance $\sigma^2 \geq 0$. Specifically,*

$$\mathbb{E}_\xi [\nabla f(x; \xi)] = \nabla f(x), \ \forall x \in \mathbb{R}^d,$$

$$\mathbb{E}_\xi \left[ \|\nabla f(x; \xi) - \nabla f(x)\|^2 \right] \leq \sigma^2, \ \forall x \in \mathbb{R}^d.$$

## 1.2. Notations

$\mathbb{R}_+ := [0, \infty)$; $\mathbb{N} := \{1, 2, \dots\}$; $\|x\|$ is the standard Euclidean norm of $x \in \mathbb{R}^d$; $\langle x, y \rangle = \sum_{i=1}^d x_i y_i$ is the standard dot product; for functions $f, g : \mathcal{Z} \to \mathbb{R}$: $g = \mathcal{O}(f)$ means that there exist $C > 0$ such that $g(z) \leq C \times f(z)$ for all $z \in \mathcal{Z}$, $g = \Omega(f)$ means that there exist $C > 0$ such that $g(z) \geq C \times f(z)$ for all $z \in \mathcal{Z}$, and $g = \Theta(f)$ means that $g = \mathcal{O}(f)$ and $g = \Omega(f)$; $[n] := \{1, 2, \dots, n\}$; $\mathbb{E}[\cdot]$ refers to mathematical expectation.

## 1.3. Related Work

Despite the variety of Asynchronous SGD algorithms proposed over the years, a fundamental question remained unresolved: *What is the optimal strategy for parallelization in this setting?*

When we have one worker, the optimal number of stochastic

---

**Input:** point $x^0 \in \mathbb{R}^d$, stepsize $\gamma > 0$, batch size $B \in \mathbb{N}$
Workers start computing stochastic gradients at $x^0$
**for** $k = 0, \dots, K - 1$ **do**
   $g_k = 0$; $b = 0$
   **while** $b < B$ **do**
      Gradient $\nabla f(x^{k-\delta^{k_b}}; \xi^{k_b})$ arrives from worker $i_{k_b}$
      **if** $\delta^{k_b} = 0$ **then**
         $g_k = g_k + \nabla f(x^{k-\delta^{k_b}}; \xi^{k_b})$; $b = b + 1$
      **end if**
      Worker $i_{k_b}$ begins calculating gradient at $x^k$
   **end while**
   Update: $x^{k+1} = x^k - \gamma \frac{g_k}{B}$
**end for**

---

gradients required to find an $\varepsilon$–stationary point is

$$\Theta \left( \frac{L\Delta}{\varepsilon} + \frac{\sigma^2 L\Delta}{\varepsilon^2} \right),$$

achieved by the vanilla SGD method (Ghadimi & Lan, 2013; Arjevani et al., 2022). In the parallel setting with many workers, several approaches have been proposed to obtain oracle lower bounds (Scaman et al., 2017; Woodworth et al., 2018; Arjevani et al., 2020; Lu & De Sa, 2021). Recent work by Tyurin & Richtárik (2023); Tyurin (2025) addressed the question by establishing lower bounds for the *time complexity* of asynchronous methods under the *fixed computation model* and the *universal computation model*. Surprisingly, they demonstrated that none of the existing Asynchronous SGD methods are optimal. Moreover, they introduced a minimax optimal method, Rennala SGD, which achieves the theoretical lower bound for time complexity.

Rennala SGD is *semi-asynchronous* and can be viewed as Minibatch SGD (which takes *synchronous* iteration/model updates) combined with an *asynchronous* minibatch collection mechanism. Let us now explain how Rennala SGD works, in the notation of Algorithm 2, which facilitates comparison with further methods described in this work. Due to the condition $\delta^{k_b} = 0$, which ignores all stochastic gradients calculated at the previous points, Rennala SGD performs the step

$$x^{k+1} = x^k - \gamma \frac{1}{B} \sum_{j=1}^B \nabla f(x^k; \xi^{k_j}),$$

where $\xi^{k_1}, \dots, \xi^{k_B}$ are independent samples from $\mathcal{D}$ collected asynchronously across all workers. Note that the workers compute the stochastic gradients at the *same* point $x^k$, with worker $i$ computing $B_i \geq 0$ gradients such that $\sum_{i=1}^n B_i = B$.

This approach has at least two fundamental drawbacks:
    (i) Once the fastest worker completes the calculation of

Table 1: The time complexities of asynchronous stochastic gradient methods, which preform the step $x^{k+1} = x^k - \gamma_k \nabla f(x^{k-\delta^k}; \xi_i^{k-\delta^k})$, to get an $\varepsilon$-stationary point in the nonconvex setting. In this table, we consider the *fixed computation model* from Section 2. Abbr.: $\sigma^2$ is defined as $\mathbb{E}_\xi[\|\nabla f(x;\xi) - \nabla f(x)\|^2] \leq \sigma^2$ for all $x \in \mathbb{R}^d$, $L$ is the smoothness constant of $f$, $\Delta := f(x^0) - f^{\inf}$, $\tau_i \in [0,\infty]$ is the time bound to compute a single stochastic gradient by worker $i$.

| Method | The Worst-Case Time Complexity Guarantees | Optimal | Adaptive to Changing Computation Times |
|---|---|---|---|
| Asynchronous SGD (Koloskova et al., 2022) (Mishchenko et al., 2022) | $\left(\frac{1}{n}\sum_{i=1}^{n}\frac{1}{\tau_i}\right)^{-1}\left(\frac{L\Delta}{\varepsilon} + \frac{\sigma^2 L\Delta}{n\varepsilon^2}\right)$ | ✘ | ✔ |
| Naive Optimal ASGD **(new)** (Algorithm 3; Theorem 2.1) | $\min_{m\in[n]}\left[\left(\frac{1}{m}\sum_{i=1}^{m}\frac{1}{\tau_i}\right)^{-1}\left(\frac{L\Delta}{\varepsilon} + \frac{\sigma^2 L\Delta}{m\varepsilon^2}\right)\right]$ | ✔ | ✘ |
| Ringmaster ASGD **(new)** (Algorithms 4 or 5; Theorem 4.2) | $\min_{m\in[n]}\left[\left(\frac{1}{m}\sum_{i=1}^{m}\frac{1}{\tau_i}\right)^{-1}\left(\frac{L\Delta}{\varepsilon} + \frac{\sigma^2 L\Delta}{m\varepsilon^2}\right)\right]$ | ✔ | ✔ |
| Lower Bound (Tyurin & Richtárik, 2023) | $\min_{m\in[n]}\left[\left(\frac{1}{m}\sum_{i=1}^{m}\frac{1}{\tau_i}\right)^{-1}\left(\frac{L\Delta}{\varepsilon} + \frac{\sigma^2 L\Delta}{m\varepsilon^2}\right)\right]$ | — | — |

the first stochastic gradient, $\nabla f(x^k; \xi^{k_1})$, it begins computing another stochastic gradient at the same point $x^k$, even though it already possesses additional information from $\nabla f(x^k; \xi^{k_1})$. Rennala SGD does not update iterate $x^k$ immediately.

(ii) Once Rennala SGD finishes the inner while loop, it will ignore all stochastic gradients that were being calculated before the loop ended, even if a worker started the calculation just a moment before. In contrast, Asynchronous SGD avoids these issues by fully utilizing all currently available information when asking a worker to calculate the next stochastic gradient and not ignoring any stochastic gradients.

These revelations raise an intriguing question: *Is asynchronous parallelization fundamentally flawed?* If the optimal solution lies in synchronous approaches, should the community abandon Asynchronous SGD and redirect its focus to developing synchronous methods? Perhaps the widespread enthusiasm for Asynchronous SGD methods was misplaced.

Alternatively, could there be a yet-to-be-discovered variant of Asynchronous SGD that achieves optimal time complexity? In this work, we answer this question affirmatively. We reestablish the prominence of Asynchronous SGD by proposing a novel asynchronous optimization method that attains optimal time complexity.

## 1.4. Contributions

Our contributions are summarized as follows:

We introduce a novel asynchronous stochastic gradient

descent method, Ringmaster ASGD, described in Algorithm 4 and Algorithm 5. This is the first asynchronous method to achieve optimal time complexity under arbitrary heterogeneous worker compute times (see Table 1). Specifically, in Theorems 4.2 and 5.1, we establish time complexities that match the lower bounds developed by Tyurin & Richtárik (2023); Tyurin (2025).

Our work begins with another new optimal method, Naive Optimal ASGD (Algorithm 3). We demonstrate that Naive Optimal ASGD achieves optimality under the fixed computation model. However, we find that it is overly simplistic and lacks robustness in scenarios where worker computation times are chaotic and dynamic. To address this limitation, we designed Ringmaster ASGD, which combines the strengths of Naive Optimal ASGD, previous non-optimal versions of Asynchronous SGD methods (Cohen et al., 2021; Koloskova et al., 2022; Mishchenko et al., 2022), and the semi-synchronous Rennala SGD (Tyurin & Richtárik, 2023).

All our claims are supported by rigorous theoretical analysis showing the optimality of the method under virtually any computation scenario, including unpredictable downtimes, fluctuations in computational performance over time, delays caused by slow or straggling workers, and challenges in maintaining synchronization across distributed systems (see Sections 4 and 5). Using numerical experiments, we demonstrate that Ringmaster ASGD outperforms existing methods (see Section G).

## 2. Preliminaries and Naive Method

To compare methods, we consider the *fixed computation model* (Mishchenko et al., 2022). In this model, it is assumed that

$$\text{worker } i \text{ takes no more than } \tau_i \text{ seconds} \tag{1}$$
$$\text{to compute a single stochastic gradient.}$$

and

$$0 < \tau_1 \le \tau_2 \le \cdots \le \tau_n, \tag{2}$$

without loss of generality. However, in Section 5, we will discuss how one can easily generalize our result to arbitrary computational dynamics, e.g., when the computation times are not bounded by the fixed values $\{\tau_i\}$, and can change in arbitrary/chaotic manner in time. Under the fixed computation model, Tyurin & Richtárik (2023) proved that the optimal time complexity lower bound is

$$T_{\text{R}} := \Theta\left(\min_{m \in [n]}\left[\left(\frac{1}{m}\sum_{i=1}^{m}\frac{1}{\tau_i}\right)^{-1}\left(\frac{L\Delta}{\varepsilon} + \frac{\sigma^2 L\Delta}{m\varepsilon^2}\right)\right]\right) \tag{3}$$

seconds achieved by Rennala SGD (Algorithm 2). However, the best analysis of Asynchronous SGD (Koloskova et al., 2022; Mishchenko et al., 2022) with appropriate stepsizes achieves the time complexity (see Sec. L in (Tyurin & Richtárik, 2023))

$$T_{\text{A}} := \Theta\left(\left(\frac{1}{n}\sum_{i=1}^{n}\frac{1}{\tau_i}\right)^{-1}\left(\frac{L\Delta}{\varepsilon} + \frac{\sigma^2 L\Delta}{n\varepsilon^2}\right)\right). \tag{4}$$

Note that $T_{\text{R}} \le T_{\text{A}}$; this is because $\min_{m \in [n]} g(m) \le g(n)$ for any function $g : \mathbb{N} \to \mathbb{R}$. Moreover, $T_{\text{R}}$ can *arbitrarily* smaller. To illustrate the difference, consider an example with $\tau_i = \sqrt{i}$ for all $i \in [n]$. Then,

$$T_{\text{R}} = \Theta\left(\max\left[\frac{\sigma L\Delta}{\varepsilon^{3/2}}, \frac{L\Delta\sigma^2}{\sqrt{n}\varepsilon^2}\right]\right)$$

and

$$T_{\text{A}} = \Theta\left(\max\left[\frac{\sqrt{n}L\Delta}{\varepsilon}, \frac{L\Delta\sigma^2}{\sqrt{n}\varepsilon^2}\right]\right),$$

see the derivations in Section E. If $n$ is large, as is often encountered in modern large-scale training scenarios, $T_{\text{A}}$ can be arbitrarily larger than $T_{\text{R}}$. Thus, the best-known variants of Asynchronous SGD are not robust to the scenarios when the number of workers is large and computation times are heterogeneous/chaotic.

### 2.1. A Naive Optimal Asynchronous SGD

We now introduce our first simple and effective strategy to improve the time complexity $T_{\text{A}}$. Specifically, we hypothesize that selecting a *subset* of workers at the beginning of the

optimization process, instead of utilizing all available workers, can lead to a more efficient and stable approach. As we shall show, this adjustment not only simplifies the computational dynamics, but also proves sufficient to achieve the optimal time complexity.

The idea is to select the fastest $[m] := \{1, 2, \ldots, m\}$ workers, thereby ignoring the slow ones and eliminating delayed gradient updates. We demonstrate that the optimal algorithm involves finding the ideal number of workers $m$ and running Asynchronous SGD (Algorithm 1) on those workers. The method is formalized in Algorithm 3.

---

**Algorithm 3** Naive Optimal ASGD

---

1: Find $m_\star \in \arg\min_{m \in [n]}\left[\left(\frac{1}{m}\sum_{i=1}^{m}\frac{1}{\tau_i}\right)^{-1}\left(1 + \frac{\sigma^2}{m\varepsilon}\right)\right]$

2: Run Asynchronous SGD (Algorithm 1) on $[m_\star]$ workers

---

The choice of $m_\star$ in Algorithm 3 effectively selects the fastest $m_\star$ workers only. Note that it is possible for $m_\star$ to be equal to $n$, meaning that all workers participate, which occurs when all workers are nearly equally fast. In this case, the harmonic mean in Line 1 of Algorithm 3 remains unchanged if all $\tau_i$s are equal, but the right-hand side decreases. However, if some workers experience large delays, the harmonic mean in Line 1 increases as $m$ grows, introducing a trade-off between the two factors. Conversely, if most workers are very slow, it may be optimal to have as few as one worker participating.

Next, we can easily prove that our algorithm, Naive Optimal ASGD (Algorithm 3), is optimal in terms of time complexity.

**Theorem 2.1.** *Consider the* fixed computation model *((1) and (2)). Let Assumptions 1.1, 1.2, and 1.3 hold. Then* Naive Optimal ASGD *(Algorithm 3) with $m_\star$ workers achieves the optimal time complexity* (3).

*Proof.* The proof is straightforward. Indeed, the time complexity (4) of Algorithm 1 with $m_*$ workers is

$$\Theta\left(\left(\frac{1}{m_*}\sum_{i=1}^{m_*}\frac{1}{\tau_i}\right)^{-1}\left(\frac{L\Delta}{\varepsilon} + \frac{\sigma^2 L\Delta}{m_*\varepsilon^2}\right)\right),$$

which equals to (3) due to the definition of $m_*$ in Algorithm 3. □

To the best of our knowledge, this is the first variant of Asynchronous SGD that provides guarantees for achieving the optimal time complexity.

**Algorithm 4** Ringmaster ASGD (without calculation stops)

**Input:** point $x^0 \in \mathbb{R}^d$, stepsize $\gamma > 0$, delay threshold $R \in \mathbb{N}$
Set $k = 0$
Workers start computing stochastic gradients at $x^0$
**while** True **do**
    Gradient $\nabla f(x^{k-\delta^k}; \xi_i^{k-\delta^k})$ arrives from worker $i$
    **if** $\delta^k < R$ **then**
        Update: $x^{k+1} = x^k - \gamma \nabla f(x^{k-\delta^k}; \xi_i^{k-\delta^k})$
        Worker $i$ begins calculating $\nabla f(x^{k+1}; \xi_i^{k+1})$
        Update the iteration number $k = k + 1$
    **else**
        Ignore the outdated gradient $\nabla f(x^{k-\delta^k}; \xi_i^{k-\delta^k})$
        Worker $i$ begins calculating $\nabla f\left(x^k; \xi_i^k\right)$
    **end if**
**end while**

**Algorithm 5** Ringmaster ASGD (with calculation stops)

**Input:** point $x^0 \in \mathbb{R}^d$, stepsize $\gamma > 0$, delay threshold $R \in \mathbb{N}$
Set $k = 0$
Workers start computing stochastic gradients at $x^0$
**while** True **do**
    Stop calculating stochastic gradients with delays $\geq R$, and start computing new ones at $x^k$ instead
    Gradient $\nabla f(x^{k-\delta^k}; \xi_i^{k-\delta^k})$ arrives from worker $i$
    Update: $x^{k+1} = x^k - \gamma \nabla f(x^{k-\delta^k}; \xi_i^{k-\delta^k})$
    Worker $i$ begins calculating $\nabla f(x^{k+1}; \xi_i^{k+1})$
    Update the iteration number $k = k + 1$
**end while**

(The core and essential modifications to Alg. 1 are highlighted in green. Alternatively, Alg. 4 is Alg. 1 with a specific choice of adaptive stepsizes defined by (5))

## 2.2. Why Is Algorithm 3 Referred to as "Naive"?

Note that determining the optimal $m_\star$ requires the knowledge of the computation times $\tau_1, \ldots, \tau_n$. If the workers' computation times were indeed static in time, this would not be an issue, as these times could be obtained by querying a single gradient from each worker before the algorithm is run. However, in real systems, computation times are rarely static, and can vary with from iteration to iteration (Dean & Barroso, 2013; Chen et al., 2016; Dutta et al., 2018; Maranjyan et al., 2025), or even become infinite at times, indicating down-time.

Naively selecting the fastest $m_*$ workers at the start of the method and keeping this selection unchanged may therefore lead to significant issues in practice. The computational environment may exhibit adversarial behavior, where worker speeds change over time. For instance, initially, the first worker may be the fastest, while the last worker is the slowest. In such cases, Naive Optimal ASGD would exclude the slowest worker. However, as time progresses, their performance may reverse, causing the initially selected $m_*$ workers, including the first worker, to become the slowest. This exposes a critical limitation of the strategy: it lacks robustness to time-varying worker speeds.

## 3. Ringmaster ASGD

We are now ready to present our new versions of Asynchronous SGD, called Ringmaster ASGD (Algorithm 4 and Algorithm 5), which guarantee the *optimal time complexity* without knowing the computation times a priori. Both methods are equivalent, up to a minor detail that we shall discuss later. Let us first focus on Algorithm 4.

## 3.1. Description

Let us compare Algorithm 4 and Algorithm 1: the only difference, highlighted with green color, lies in the fact that Algorithm 4 disregards "very old stochastic gradients," which are gradients computed at points with significant delay. Specifically, Algorithm 4 receives $\nabla f(x^{k-\delta^k}; \xi_i^{k-\delta^k})$, compares $\delta^k$ to our *delay threshold* hyperparameter $R$. If $\delta^k \geq R$, then Algorithm 4 completely ignores this very outdated stochastic gradient and requests the worker to compute a new stochastic gradient at the most relevant point $x^k$.

Notice that Ringmaster ASGD (Algorithm 5) is Algorithm 1 with the following adaptive stepsize rule:

$$\gamma_k = \begin{cases} \gamma, & \text{if} \quad \bar{\delta}_i^k < R, \\ 0, & \text{if} \quad \bar{\delta}_i^k \geq R, \end{cases}$$

$$\bar{\delta}_j^{k+1} = \begin{cases} 0, & \text{if} \quad j = i, \\ \bar{\delta}_j^k + 1, & \text{if} \quad j \neq i \quad \& \quad \bar{\delta}_i^k < R, \\ \bar{\delta}_j^k, & \text{if} \quad j \neq i \quad \& \quad \bar{\delta}_i^k \geq R, \end{cases} \quad (5)$$

where $i$ is the index of the worker whose stochastic gradient is applied at iteration $k$, and where we initialize the *virtual sequence of delays* $\{\bar{\delta}_j^k\}_{k,j}$ by setting $\bar{\delta}_j^0 = 0$ for all $j \in [n]$.

## 3.2. Delay Threshold

Returning to Algorithm 4, note that we have a parameter $R$ called the *delay threshold*. When $R = 1$, the algorithm reduces to the classical SGD method, i.e.,

$$x^{k+1} = x^k - \gamma \nabla f(x^k; \xi_i^k),$$

since $\delta^k = 0$ for all $k \geq 0$. In this case, the algorithm becomes highly conservative, ignoring all stochastic gradi-

ents computed at earlier points $x^{k-1}, \ldots, x^0$. Conversely, if $R = \infty$, the method incorporates stochastic gradients with arbitrarily large delays, and becomes classical Asynchronous SGD. Intuitively, there should be a balance – a "proper" value of $R$ that would i) prevent the method from being overly conservative, while ii) ensuring stability by making sure that only informative stochastic gradients are used to update the model. We formalize these intuitions by proposing an optimal $R$ in Section 4. Interestingly, the value of $R$ does *not* depend on the computation times.

### 3.3. Why Do We Ignore the Old Stochastic Gradients?

The primary reason is that doing so enables the development of the first optimal Asynchronous SGD method that achieves the lower bounds (see Sections 4 and 5). Ignoring old gradients allows us to establish tighter convergence guarantees. Intuitively, old gradients not only fail to provide additional useful information about the function $f$, but they can also negatively impact the algorithm's performance. Therefore, disregarding them in the optimization process is essential for achieving our goal of developing an optimal Asynchronous SGD method.

### 3.4. Comparison to Rennala SGD

Unlike Rennala SGD, which combines a synchronous Minibatch SGD update with an asynchronous minibatch collection strategy, Ringmaster ASGD is fully asynchronous, which, as we show in our experiments, provided further practical advantages:

(i) Ringmaster ASGD updates the model immediately upon receiving a new and relevant (i.e., not too outdated) stochastic gradient. This immediate update strategy is particularly advantageous for sparse models in practice, where different gradients may update disjoint parts of the model only, facilitating faster processing. This concept aligns with the ideas of Recht et al. (2011), where lock-free asynchronous updates were shown to effectively leverage sparsity for improved performance.

(ii) Ringmaster ASGD ensures that stochastic gradients computed by fast workers are never ignored. In contrast, Rennala SGD may discard stochastic gradients, even if they were recently initiated and would be computed quickly (see discussion in Section 1).

At the same time, Ringmaster ASGD adheres to the same principles that make Rennala SGD optimal. The core philosophy of Rennala SGD is to prioritize fast workers by employing asynchronous batch collection, effectively ignoring slow workers. This is achieved by carefully choosing the batch size $B$ in Algorithm 2: large enough to allow fast workers to complete their calculations, but small enough to disregard slow workers and excessively delayed stochastic gradients. Similarly, Ringmaster ASGD implements this

concept using the delay threshold $R$.

In summary, while both Rennala SGD and Ringmaster ASGD are optimal from a theoretical perspective, the complete absence of synchronization in the latter method intuitively makes it more appealing in practice, and allows it to collect further gains which are not captured in theory. As we shall see, this intuition is supported by our experiments.

### 3.5. Comparison to Previous Asynchronous SGD Variants

Koloskova et al. (2022) and Mishchenko et al. (2022) provide the previous state-of-the-art analysis of Asynchronous SGD. However, as demonstrated in Section 2, their analysis does not guarantee optimality. This is due to at least two factors:

(i) their approaches do *not* discard old stochastic gradients, instead attempting to utilize all gradients, even those with very large delays;

(ii) although they select stepsizes $\gamma_k$ that decrease as the delays increase, this adjustment may not be sufficient to ensure optimal performance, and the choice of their stepsize may be suboptimal compared to (5).

### 3.6. Stopping the Irrelevant Computations

If stopping computations is feasible, we can further enhance Algorithm 4 by introducing Algorithm 5. Instead of waiting for workers to complete the calculation of outdated stochastic gradients with delays larger than $R$ which would not be used anyway, we propose to *stop/terminate* these computations immediately and reassign the workers to the most relevant point $x^k$. This adjustment provides workers with an opportunity to catch up, as they may become faster and perform computations more efficiently at the updated point.

## 4. Theoretical Analysis

We are ready to present the theoretical analysis of Ringmaster ASGD. We start with an *iteration complexity* bound:

**Theorem 4.1** (Proof in Appendix C). *Under Assumptions 1.1, 1.2, and 1.3, let the stepsize in* Ringmaster ASGD *(Algorithm 4 or Algorithm 5) be*

$$\gamma = \min \left\{ \frac{1}{2RL}, \frac{\varepsilon}{4L\sigma^2} \right\}.$$

*Then*

$$\frac{1}{K+1} \sum_{k=0}^{K} \mathbb{E}\left[\left\|\nabla f\left(x^k\right)\right\|^2\right] \leq \varepsilon,$$

*as long as*

$$K \geq \frac{8RL\Delta}{\epsilon} + \frac{16\sigma^2 L\Delta}{\epsilon^2}, \tag{6}$$

*where $R \in \{1, 2, \ldots, \}$ is an arbitrary delay threshold.*

The classical analysis of Asynchronous SGD achieves the same convergence rate, with $R$ defined as $R \equiv \max_{k \in [K]} \delta^k$ (Stich & Karimireddy, 2020; Arjevani et al., 2020). This outcome is expected, as setting $R = \max_{k \in [K]} \delta^k$ in Ringmaster ASGD makes it equivalent to classical Asynchronous SGD, since no gradients are ignored. Furthermore, the analyses by Cohen et al. (2021); Koloskova et al. (2022); Mishchenko et al. (2022) yield the same rate with $R = n$. However, it is important to note that $R$ is a free parameter in Ringmaster ASGD that can be chosen arbitrarily. While setting $R = \max_{k \in [K]} \delta^k$ or $R = n$ effectively recovers the earlier *iteration complexities*, we show that there exists a *different* choice of $R$ leading to optimal time complexities (see Theorems 4.2 and 5.1).

It is important to recognize that the *iteration complexity* (6) does *not* capture the actual "runtime" performance of the algorithm. To select an optimal value for $R$, we must shift our focus from *iteration complexity* to *time complexity*, which measures the algorithm's runtime. We achieve the best practical and effective choice by optimizing the *time complexity* over $R$. In order to find the *time complexity*, we need the following lemma.

**Lemma 4.1** (Proof in Appendix A). *Let the workers' computation times satisfy the* fixed computation model *((1) and (2)). Let $R$ be the delay threshold of Algorithm 4 or Algorithm 5. The time required to complete any $R$ consecutive iterate updates of Algorithm 4 or Algorithm 5 is at most*

$$
t(R) := 2 \min_{m \in [n]} \left[ \left( \frac{1}{m} \sum_{i=1}^{m} \frac{1}{\tau_i} \right)^{-1} \left( 1 + \frac{R}{m} \right) \right]. \quad (7)
$$

Combining Theorem 4.1 and Lemma 4.1, we provide our main result.

**Theorem 4.2** (Optimality of Ringmaster ASGD). *Let Assumptions 1.1, 1.2, and 1.3 hold. Let the stepsize in* Ringmaster ASGD *(Algorithm 4 or Algorithm 5) be $\gamma = \min \left\{ \frac{1}{2RL}, \frac{\varepsilon}{4L\sigma^2} \right\}$. Then, under the* fixed computation model *((1) and (2)),* Ringmaster ASGD *achieves the optimal time complexity*

$$
\mathcal{O} \left( \min_{m \in [n]} \left[ \left( \frac{1}{m} \sum_{i=1}^{m} \frac{1}{\tau_i} \right)^{-1} \left( \frac{L\Delta}{\varepsilon} + \frac{\sigma^2 L\Delta}{m\varepsilon^2} \right) \right] \right) \quad (8)
$$

*with the delay threshold*

$$
R = \max \left\{ 1, \left\lceil \frac{\sigma^2}{\varepsilon} \right\rceil \right\}. \quad (9)
$$

Note that the value of $R$ does not in any way depend on the computation times $\{\tau_1, \ldots, \tau_n\}$.

*Proof.* From Theorem 4.1, the iteration complexity of Ringmaster ASGD is

$$
K = \left\lceil \frac{8RL\Delta}{\epsilon} + \frac{16\sigma^2 L\Delta}{\epsilon^2} \right\rceil. \quad (10)
$$

Using Lemma 4.1, we know that Ringmaster ASGD requires at most $t(R)$ seconds to finish any $R$ consecutive updates of the iterates. Therefore, the total time is at most

$$
t(R) \times \left\lceil \frac{K}{R} \right\rceil.
$$

Without loss of generality, we assume[2] that $L\Delta > \varepsilon/2$. Therefore, using (10), we get

$$
t(R) \times \left\lceil \frac{K}{R} \right\rceil = \mathcal{O} \left( t(R) \left( \frac{L\Delta}{\epsilon} + \frac{\sigma^2 L\Delta}{R\epsilon^2} \right) \right). \quad (11)
$$

It is left to substitute our choice (9) into (11) to get (8). We take (9), noticing that this choice of $R$ minimizes (11) up to a universal constant. $\square$

To the best of our knowledge, this is the first result in the literature to establish the optimality of a fully asynchronous variant of SGD: the time complexity (8) is optimal and aligns with the lower bound established by Tyurin & Richtárik (2023).

The derived time complexity (8) has many nice and desirable properties. First, it is robust to slow workers: if $\tau_n \to \infty$, the expression equals

$$
\min_{m \in [n-1]} \left( \frac{1}{m} \sum_{i=1}^{m} \frac{1}{\tau_i} \right)^{-1} \left( \frac{L\Delta}{\varepsilon} + \frac{\sigma^2 L\Delta}{m\varepsilon^2} \right),
$$

effectively disregarding the slowest worker. Next, assume that $m_*$ is the smallest index that minimizes (8). In this case, (8) simplifies further to

$$
\left( \frac{1}{m_*} \sum_{i=1}^{m_*} \frac{1}{\tau_i} \right)^{-1} \left( \frac{L\Delta}{\varepsilon} + \frac{\sigma^2 L\Delta}{m_*\varepsilon^2} \right).
$$

This shows that the method operates effectively as if only the fastest $m_*$ workers participate in the optimization process, resembling the idea from Algorithm 3. What is important, however, the method determines $m_*$ adaptively and automatically.

### 4.1. The Choice of Threshold

Another notable property of the method and the time complexity result in Theorem 4.2 is that the threshold $R$ is independent of the individual computation times $\tau_i$. As a result,

---

[2] Otherwise, using $L$–smoothness, $\left\| \nabla f(x^0) \right\|^2 \leq 2L\Delta \leq \varepsilon$, and the initial point is an $\varepsilon$–stationary point. See Section F.

the method can be applied across heterogeneous distributed systems—where workers have varying speeds—while still achieving the optimal time complexity given in (11), up to a constant factor.

If a tighter, constant-level expression for the optimal threshold is desired, $R$ can be computed explicitly. In that case, it will depend on the values of $\tau_i$. The optimal $R$ solves

$$\arg \min_{R \geq 1} \left\{ t(R) \left( 1 + \frac{\sigma^2}{R\varepsilon} \right) \right\},$$

which follows from (11) after discarding constants independent of $R$. Here, $t(R)$ denotes the total time required for $R$ consecutive iterations, and is upper bounded by (7).

Substituting this bound yields the following expression for the optimal $R$:

$$R = \max \left\{ \sigma \sqrt{\frac{m^*}{\varepsilon}}, 1 \right\},$$

where

$$m^* = \arg \min_{m \in [n]} \left\{ \left( \frac{1}{m} \sum_{i=1}^{m} \frac{1}{\tau_i} \right)^{-1} \left( 1 + 2\sqrt{\frac{\sigma^2}{m\varepsilon}} + \frac{\sigma^2}{m\varepsilon} \right) \right\}.$$

As the expression shows, the optimal threshold $R$ depends on $m^*$, which in turn is determined by the $\tau_i$ values.

## 4.2. Proof Techniques

Several mathematical challenges had to be addressed to achieve the final result. Lemmas 4.1 and 5.1 are novel, as estimating the time complexity of the *asynchronous* Ringmaster ASGD method requires distinct approaches compared to the *semi-synchronous* Rennala SGD method because Ringmaster ASGD is more chaotic and less predictable. Compared to (Koloskova et al., 2022; Mishchenko et al., 2022), the proof of Theorem 4.1 is tighter and more refined, as we more carefully analyze the sum

$$\sum_{k=0}^{K} \mathbb{E} \left[ \left\| x^k - x^{k-\delta^k} \right\|^2 \right]$$

in Lemma C.2. We believe that the simplicity of Ringmaster ASGD, combined with the new lemmas, the refined analysis, and the novel choice of $R$ in Theorem 4.2, represents a set of non-trivial advancements that enable us to achieve the optimal time complexity.

# 5. Optimality Under Arbitrary Computation Dynamics

In the previous sections, we presented the motivation, improvements, comparisons, and theoretical results within the

framework of the *fixed computation model* ((1) and (2)). We now prove Ringmaster ASGD is optimal under virtually any computation behavior of the workers. One way to formalize these behaviors is to use the *universal computation model* (Tyurin, 2025).

For each worker $i \in [n]$, we associate a *computation power* function $v_i : \mathbb{R}_+ \to \mathbb{R}_+$. The number of stochastic gradients that worker $i$ can compute between times $T_0$ and $T_1$ is given by the integral of its computation power $v_i$, followed by applying the floor operation:

$$\text{"\# of stoch. grad. in } [T_0, T_1]\text{"} = \left\lfloor \int_{T_0}^{T_1} v_i(\tau)d\tau \right\rfloor. \quad (12)$$

The computational power $v_i$ characterizes the behavior of workers, accounting for potential disconnections due to hardware or network delays, variations in processing capacity over time, and fluctuations or trends in computation speeds. The integral of $v_i$ over a given interval represents the *computation work* performed. If $v_i$ is small within $[T_0, T_1]$, the integral is correspondingly small, indicating that worker $i$ performs less computation. Conversely, if $v_i$ is large over $[T_0, T_1]$, the worker is capable of performing more computation.

For our analysis, we only assume that $v_i$ is non-negative and continuous almost everywhere[3]. We use this assumption non-explicitly when applying the Riemann integral. The computational power $v_i$ can even vary randomly, and all the results discussed hold conditional on the randomness of $\{v_i\}$.

Let us examine some examples. If worker $i$ remains inactive for the first $t$ seconds and then becomes active, this corresponds to $v_i(\tau) = 0$ for all $\tau \leq t$ and $v_i(\tau) > 0$ for all $\tau > t$. Furthermore, we allow $v_i$ to exhibit periodic or even chaotic behavior[4]. The *universal computation model* reduces to the *fixed computation model* when $v_i(t) = 1/\tau_i$ for all $t \geq 0$ and $i \in [n]$. In this case, $(12) = \lfloor T_1 - T_0/\tau_i \rfloor$, indicating that worker $i$ computes one stochastic gradient after $T_0 + \tau_i$ seconds, two stochastic gradients after $T_0 + 2\tau_i$ seconds, and so forth.

Note that Theorem 4.1 is valid under any computation model. Next, we introduce an alternative to Lemma 4.1 for the *universal computation model*.

**Lemma 5.1** (Proof in Appendix B). *Let the workers' computation times satisfy the universal computation model. Let $R$ be the delay threshold of Algorithm 4 or Algorithm 5. Assume that some iteration starts at time $T_0$. Starting from this iteration, the $R$ consecutive iterate updates of Algorithm 4*

---

[3]Thus, it can be discontinuous and "jump" on a countable set.

[4]For example, $v_i(t)$ might behave discontinuously as follows: $v_i(t) = 0.5t + \sin(10t)$ for $t \leq 10$, $v_i(t) = 0$ for $10 < t \leq 20$, and $v_i(t) = \max(80 - 0.5t, 0)$.

*or Algorithm 5 will be performed before the time*

$$T(R, T_0) := \min\left\{T \geq 0 : \sum_{i=1}^{n} \left\lfloor \frac{1}{4} \int_{T_0}^{T} v_i(\tau)d\tau \right\rfloor \geq R\right\}.$$

This result extends Lemma 4.1. Specifically, if $v_i(t) = 1/\tau_i$ for all $t \geq 0$ and $i \in [n]$, it can be shown that $T(R, T_0) - T_0 = \Theta(t(R))$ for all $T_0 \geq 0$.

Combining Theorem 4.1 and Lemma 5.1, we are ready to present our main result.

**Theorem 5.1** (Optimality of Ringmaster ASGD; Proof in Appendix D). *Let Assumptions 1.1, 1.2, and 1.3 hold. Let the stepsize in* Ringmaster ASGD *(Algorithm 4 or Algorithm 5) be*

$$\gamma = \min\left\{\frac{1}{2RL}, \frac{\varepsilon}{4L\sigma^2}\right\},$$

*and delay threshold*

$$R = \max\left\{1, \left\lceil\frac{\sigma^2}{\varepsilon}\right\rceil\right\}.$$

*Then, under the* universal computation model, Ringmaster ASGD *finds an $\varepsilon$–stationary point after at most $T_{\bar{K}}$ seconds, where $\bar{K} := \left\lceil\frac{48L\Delta}{\epsilon}\right\rceil$ and $T_{\bar{K}}$ is the $\bar{K}^{th}$ element of the following recursively defined sequence:*

$$T_K := \min\left\{T \geq 0 : \sum_{i=1}^{n} \left\lfloor \frac{1}{4} \int_{T_{K-1}}^{T} v_i(\tau)d\tau \right\rfloor \geq R\right\}$$

*for all $K \geq 1$ and $T_0 = 0$.*

Granted, the obtained theorem is less explicit than Theorem 4.2. However, this is the price to pay for the generality of the result in terms of the computation time dynamics it allows. Determining the time complexity $T_{\bar{K}}$ requires computing $T_1, T_2$, and so on, sequentially. However, in certain scenarios, $T_{\bar{K}}$ can be derived explicitly. For example, if $v_i(t) = 1/\tau_i$ for all $t \geq 0$ and $i \in [n]$, then $T_{\bar{K}}$ is given by (8). Furthermore, $T_1$ represents the number of seconds required to compute the first $R$ stochastic gradients, $T_2$ represents the time needed to compute the first $2 \times R$ stochastic gradients, and so on. Naturally, to compute $T_2$, one must first determine $T_1$, which is reasonable given the sequential nature of the process.

The obtained result is optimal, aligns with the lower bound established by Tyurin (2025), and cannot be improved by any asynchronous parallel method.

## 6. Conclusion and Future Work

In this work, we developed the first Asynchronous SGD method, named Ringmaster ASGD, that achieves optimal time complexity. By employing a carefully designed algorithmic approach, which can be interpreted as Asynchronous SGD with adaptive step sizes (5), we successfully reach this goal. By selecting an appropriate delay threshold $R$ in Algorithm 4, the method attains the theoretical lower bounds established by Tyurin & Richtárik (2023); Tyurin (2025).

Future work can explore heterogeneous scenarios, where the data on each device comes from different distributions, which are particularly relevant for federated learning (FL) (Konečný et al., 2016; McMahan et al., 2016; Kairouz et al., 2021). In FL, communication costs also become crucial, making it worthwhile to consider similar extensions (Alistarh et al., 2017; Tyurin et al., 2024b; Tyurin & Richtárik, 2024). It would be also interesting to design a fully asynchronous analog to the method from (Tyurin et al., 2024a). Additionally, as discussed by Maranjyan et al. (2025), computation and communication times can be treated as random variables. It would be valuable to investigate these cases further and derive closed-form expressions for various distributions.

## Acknowledgments

The research reported in this publication was supported by funding from King Abdullah University of Science and Technology (KAUST): i) KAUST Baseline Research Scheme, ii) Center of Excellence for Generative AI, under award number 5940, iii) SDAIA-KAUST Center of Excellence in Artificial Intelligence and Data Science. The work of A.T. was supported by the Ministry of Economic Development of the Russian Federation (code 25-139-66879-1-0003). We would like to thank Adrien Fradin for valuable discussions and helpful suggestions that contributed to the improvement of this work.

## Impact Statement

This paper presents work whose goal is to advance the field of Machine Learning. There are many potential societal consequences of our work, none which we feel must be specifically highlighted here.

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

## A. Proof of Lemma 4.1

**Lemma 4.1.** *Let workers' computation times satisfy the* fixed computation model *((1) and (2)). Let $R$ be the delay threshold of Algorithm 4 or Algorithm 5. The time required to complete any $R$ consecutive iterates updates of Algorithm 4 or Algorithm 5 is at most*

$$t(R) := 2 \min_{m \in [n]} \left[ \left( \frac{1}{m} \sum_{i=1}^{m} \frac{1}{\tau_i} \right)^{-1} \left( 1 + \frac{R}{m} \right) \right]. \tag{7}$$

*Proof.* We now focus on Algorithm 5. Let us consider a simplified notation of $t(R)$:

$$t := t(R) = 2 \min_{m \in [n]} \left( \left( \sum_{i=1}^{m} \frac{1}{\tau_i} \right)^{-1} (R + m) \right) = 2 \min_{m \in [n]} \left( \frac{1}{m} \sum_{i=1}^{m} \frac{1}{\tau_i} \right)^{-1} \left( 1 + \frac{R}{m} \right).$$

Let us fix any iteration $k$, and consider the consecutive iterations from $k$ to $k + R - 1$. By the design of Algorithm 5, any worker will be stopped at most one time, meaning that at most one stochastic gradient will be ignored from the worker. After that, the delays of the next stochastic gradients will not exceed $R - 1$ during these iterations. As soon as some worker finishes calculating a stochastic gradient, it immediately starts computing a new stochastic gradient.

Using a proof by contradiction, assume that Algorithm 5 will not be able to finish iteration $k + R - 1$ by the time $t$. At the same time, by the time $t$, all workers will calculate at least

$$\sum_{i=1}^{n} \max \left\{ \left\lfloor \frac{t}{\tau_i} \right\rfloor - 1, 0 \right\} \tag{13}$$

stochastic gradients with delays $< R$ because $\left\lfloor \frac{t}{\tau_i} \right\rfloor$ is the number of stochastic gradients that worker $i$ can calculate after $t$ seconds. We subtract 1 to account for the fact that one stochastic gradient with a large delay can be ignored. Let us take an index

$$j^* = \arg \min_{m \in [n]} \left( \left( \sum_{i=1}^{m} \frac{1}{\tau_i} \right)^{-1} (R + m) \right).$$

Since $\lfloor x \rfloor \geq x - 1$ for all $x \geq 0$, we get

$$\sum_{i=1}^{n} \max \left\{ \left\lfloor \frac{t}{\tau_i} \right\rfloor - 1, 0 \right\} \geq \sum_{i=1}^{j^*} \max \left\{ \left\lfloor \frac{t}{\tau_i} \right\rfloor - 1, 0 \right\} \geq \sum_{i=1}^{j^*} \left( \left\lfloor \frac{t}{\tau_i} \right\rfloor - 1 \right) \geq \sum_{i=1}^{j^*} \frac{t}{\tau_i} - 2j^*$$

$$= 2 \left( \sum_{i=1}^{j^*} \frac{1}{\tau_i} \right) \left( \left( \sum_{i=1}^{j^*} \frac{1}{\tau_i} \right)^{-1} (R + j^*) \right) - 2j^* = 2R + 2j^* - 2j^* \geq R.$$

We can conclude that by the time (7), the algorithm will calculate $R$ stochastic gradients with delays $< R$ and finish iteration $k + R - 1$, which contradicts the assumption.

The proof for Algorithm 4 is essentially the same. In Algorithm 4, one stochastic gradient with a large delay for each worker will be ignored, and all other stochastic will be used in the optimization process. □

## B. Proof of Lemma 5.1

**Lemma 5.1.** *Let the workers' computation times satisfy the* universal computation model. *Let $R$ be the delay threshold of Algorithm 4 or Algorithm 5. Assume that some iteration starts at time $T_0$. Starting from this iteration, the $R$ consecutive iterative updates of Algorithm 4 or Algorithm 5 will be performed before the time*

$$T(R, T_0) := \min \left\{ T \geq 0 : \sum_{i=1}^{n} \left\lfloor \frac{1}{4} \int_{T_0}^{T} v_i(\tau) d\tau \right\rfloor \geq R \right\}.$$

*Proof.* Let

$$T := T(R, T_0) = \min\left\{T \geq 0 : \sum_{i=1}^{n} \left\lfloor \frac{1}{4} \int_{T_0}^{T} v_i(\tau)d\tau \right\rfloor \geq R\right\}.$$

At the beginning, this proof follows the proof of Lemma 4.1 up to (13). Here we also use a proof by contradiction and assume that Algorithm 5 will not be able to do $R$ consecutive iterative updates by the time $T$.

Instead of (13), all workers will calculate at least

$$N := \sum_{i=1}^{n} \max\left\{\left\lfloor \int_{T_0}^{T} v_i(\tau)d\tau \right\rfloor - 1, 0\right\}$$

stochastic gradients by time $T$ because, due to (12), $\left\lfloor \int_{T_0}^{T} v_i(\tau)d\tau \right\rfloor$ is the number of stochastic gradients that worker $i$ will calculate in the interval $[T_0, T]$. We define $V_i(T) := \int_{T_0}^{T} v_i(\tau)d\tau$. Thus,

$$N = \sum_{i=1}^{n} \max\left\{\lfloor V_i(T) \rfloor - 1, 0\right\} \text{ and } T = \min\left\{T \geq 0 : \sum_{i=1}^{n} \left\lfloor \frac{V_i(T)}{4} \right\rfloor \geq R\right\}.$$

Let us additionally define

$$S := \left\{i \in [n] : \frac{V_i(T)}{4} \geq 1\right\}. \tag{14}$$

Note that

$$\sum_{i=1}^{n} \left\lfloor \frac{V_i(T)}{4} \right\rfloor = \sum_{i \in S} \left\lfloor \frac{V_i(T)}{4} \right\rfloor, \tag{15}$$

since $\left\lfloor \frac{V_i(T)}{4} \right\rfloor = 0$ for all $i \notin S$. Using simple algebra, we get

$$N = \sum_{i=1}^{n} \max\{\lfloor V_i(T) \rfloor - 1, 0\} \geq \sum_{i \in S} \max\{\lfloor V_i(T) \rfloor - 1, 0\} \geq \sum_{i \in S} \lfloor V_i(T) \rfloor - |S| \geq \sum_{i \in S} V_i(T) - 2|S|,$$

where the last inequality due to $\lfloor x \rfloor \geq x - 1$ for all $x \in \mathbb{R}$. Next

$$N \overset{(14)}{\geq} \frac{1}{2}\sum_{i \in S} V_i(T) + \frac{1}{2}\sum_{i \in S} 4 - 2|S| = \sum_{i \in S} \frac{V_i(T)}{2} \geq \sum_{i \in S} \left\lfloor \frac{V_i(T)}{4} \right\rfloor \overset{(15)}{=} \sum_{i=1}^{n} \left\lfloor \frac{V_i(T)}{4} \right\rfloor \geq R,$$

where the last inequality due to the definition of $T$. As in the proof of Lemma 4.1, we can conclude that by the time $T = T(R, T_0)$, the algorithm will calculate $R$ stochastic gradients with delays $< R$ and finish iteration $k + R - 1$, which contradicts the assumption. $\qquad\square$

## C. Proof of Theorem 4.1

**Theorem 4.1.** *Under Assumptions 1.1, 1.2, and 1.3, let the stepsize in* Ringmaster ASGD *(Algorithm 4 or Algorithm 5) be*

$$\gamma = \min\left\{\frac{1}{2RL}, \frac{\varepsilon}{4L\sigma^2}\right\}.$$

*Then, the following holds*

$$\frac{1}{K+1}\sum_{k=0}^{K} \mathbb{E}\left[\left\|\nabla f\left(x^k\right)\right\|^2\right] \leq \varepsilon,$$

*for*

$$K \geq \frac{8RL\Delta}{\epsilon} + \frac{16\sigma^2 L\Delta}{\epsilon^2}, \tag{6}$$

*where $R \in \{1, 2, \ldots, \}$ is an arbitrary delay threshold.*

We adopt the proof strategy from Koloskova et al. (2022) and rely on the following two lemmas.

**Lemma C.1** (Descent Lemma; Proof in Appendix C.1). *Under Assumptions 1.1 and 1.3, if the stepsize in* Ringmaster ASGD *(Algorithm 4 or Algorithm 5) satisfies* $\gamma \leq \frac{1}{2L}$*, the following inequality holds:*

$$\mathbb{E}_{k+1}\left[f\left(x^{k+1}\right)\right] \leq f\left(x^k\right) - \frac{\gamma}{2}\left\|\nabla f\left(x^k\right)\right\|^2 - \frac{\gamma}{4}\left\|\nabla f\left(x^{k-\delta^k}\right)\right\|^2 + \frac{\gamma L^2}{2}\left\|x^k - x^{k-\delta^k}\right\|^2 + \frac{\gamma^2 L}{2}\sigma^2,$$

*where* $\mathbb{E}_{k+1}\left[\cdot\right]$ *represents the expectation conditioned on all randomness up to iteration* $k$*.*

**Lemma C.2** (Residual Estimation; Proof in Appendix C.2). *Under Assumptions 1.1 and 1.3, the iterates of* Ringmaster ASGD *(Algorithm 4 or Algorithm 5) with stepsize* $\gamma \leq \frac{1}{2RL}$ *satisfy the following bound:*

$$\frac{1}{K+1}\sum_{k=0}^{K}\mathbb{E}\left[\left\|x^k - x^{k-\delta^k}\right\|^2\right] \leq \frac{1}{2L^2(K+1)}\sum_{k=0}^{K}\mathbb{E}\left[\left\|\nabla f\left(x^{k-\delta^k}\right)\right\|^2\right] + \frac{\gamma}{L}\sigma^2.$$

With these results, we are now ready to prove Theorem 4.1.

*Proof of Theorem 4.1.* We begin by averaging over $K+1$ iterations and dividing by $\gamma$ in the inequality from Lemma C.1:

$$\frac{1}{K+1}\sum_{k=0}^{K}\left(\frac{1}{2}\mathbb{E}\left[\left\|\nabla f\left(x^k\right)\right\|^2\right] + \frac{1}{4}\mathbb{E}\left[\left\|\nabla f(x^{k-\delta^k})\right\|^2\right]\right) \leq \frac{\Delta}{\gamma(K+1)} + \frac{\gamma L}{2}\sigma^2$$
$$+ \frac{1}{K+1}\frac{L^2}{2}\sum_{k=0}^{K}\mathbb{E}\left[\left\|x^k - x^{k-\delta^k}\right\|^2\right].$$

Next, applying Lemma C.2 to the last term, we have:

$$\frac{1}{K+1}\sum_{k=0}^{K}\left(\frac{1}{2}\mathbb{E}\left[\left\|\nabla f\left(x^k\right)\right\|^2\right] + \frac{1}{4}\mathbb{E}\left[\left\|\nabla f(x^{k-\delta^k})\right\|^2\right]\right) \leq \frac{\Delta}{\gamma(K+1)} + \frac{\gamma L}{2}\sigma^2$$
$$+ \frac{1}{4(K+1)}\sum_{k=0}^{K}\mathbb{E}\left[\left\|\nabla f\left(x^{k-\delta^k}\right)\right\|^2\right] + \frac{\gamma L}{2}\sigma^2.$$

Simplifying further, we obtain:

$$\frac{1}{K+1}\sum_{k=0}^{K}\mathbb{E}\left[\left\|\nabla f\left(x^k\right)\right\|^2\right] \leq \frac{2\Delta}{\gamma(K+1)} + 2\gamma L\sigma^2.$$

Now, we choose the stepsize $\gamma$ as:

$$\gamma = \min\left\{\frac{1}{2RL}, \frac{\varepsilon}{4L\sigma^2}\right\} \leq \frac{1}{2RL}.$$

With this choice of $\gamma$, it remains to choose

$$K \geq \frac{8\Delta RL}{\epsilon} + \frac{16\Delta L\sigma^2}{\epsilon^2}$$

to ensure

$$\frac{1}{K+1}\sum_{k=0}^{K}\mathbb{E}\left[\left\|\nabla f\left(x^k\right)\right\|^2\right] \leq \epsilon.$$

This completes the proof.

$\square$

## C.1. Proof of Lemma C.1

**Lemma C.1** (Descent Lemma). *Under Assumptions 1.1 and 1.3, if the stepsize in* Ringmaster ASGD *(Algorithm 4 or Algorithm 5) satisfies $\gamma \leq \frac{1}{2L}$, the following inequality holds:*

$$\mathbb{E}_{k+1}\left[f\left(x^{k+1}\right)\right] \leq f\left(x^k\right) - \frac{\gamma}{2}\left\|\nabla f\left(x^k\right)\right\|^2 - \frac{\gamma}{4}\left\|\nabla f\left(x^{k-\delta^k}\right)\right\|^2 + \frac{\gamma L^2}{2}\left\|x^k - x^{k-\delta^k}\right\|^2 + \frac{\gamma^2 L}{2}\sigma^2,$$

*where $\mathbb{E}_{k+1}[\cdot]$ represents the expectation conditioned on all randomness up to iteration $k$.*

*Proof.* Assume that we get a stochastic gradient from the worker with index $i_k$ when calculating $x^{k+1}$. Since the function $f$ is $L$-smooth (Assumption 1.1), we have (Nesterov, 2018):

$$\mathbb{E}_{k+1}\left[f\left(x^{k+1}\right)\right] \leq f\left(x^k\right) - \gamma \underbrace{\mathbb{E}_{k+1}\left[\left\langle\nabla f\left(x^k\right), \nabla f\left(x^{k-\delta^k}; \xi_{i_k}^{k-\delta^k}\right)\right\rangle\right]}_{=:t_1} + \frac{L}{2}\gamma^2 \underbrace{\mathbb{E}_{k+1}\left[\left\|\nabla f\left(x^{k-\delta^k}; \xi_{i_k}^{k-\delta^k}\right)\right\|^2\right]}_{=:t_2}.$$

Using Assumption 1.3, we estimate the second term as

$$t_1 = \left\langle\nabla f\left(x^k\right), \nabla f\left(x^{k-\delta^k}\right)\right\rangle = \frac{1}{2}\left[\left\|\nabla f\left(x^k\right)\right\|^2 + \left\|\nabla f\left(x^{k-\delta^k}\right)\right\|^2 - \left\|\nabla f\left(x^k\right) - \nabla f\left(x^{k-\delta^k}\right)\right\|^2\right].$$

Using the variance decomposition equality and Assumption 1.3, we get

$$\begin{aligned}
t_2 &= \mathbb{E}_{k+1}\left[\left\|\nabla f\left(x^{k-\delta^k}; \xi_{i_k}^{k-\delta^k}\right) - \nabla f\left(x^{k-\delta^k}\right)\right\|^2\right] + \left\|\nabla f\left(x^{k-\delta^k}\right)\right\|^2 \\
&\leq \sigma^2 + \left\|\nabla f\left(x^{k-\delta^k}\right)\right\|^2.
\end{aligned}$$

Combining the results for $t_1$ and $t_2$, and using $L$-smoothness to bound $\left\|\nabla f\left(x^k\right) - \nabla f\left(x^{k-\delta^k}\right)\right\|^2$, we get:

$$\mathbb{E}_{k+1}\left[f\left(x^{k+1}\right)\right] \leq f\left(x^k\right) - \frac{\gamma}{2}\left\|\nabla f\left(x^k\right)\right\|^2 - \frac{\gamma}{2}(1-\gamma L)\left\|\nabla f\left(x^{k-\delta^k}\right)\right\|^2 + \frac{\gamma L^2}{2}\left\|x^k - x^{k-\delta^k}\right\|^2 + \frac{\gamma^2 L}{2}\sigma^2.$$

Finally, applying the condition $\gamma \leq \frac{1}{2L}$ completes the proof. $\qquad\square$

## C.2. Proof of Lemma C.2

**Lemma C.2** (Residual Estimation). *Under Assumptions 1.1 and 1.3, the iterates of* Ringmaster ASGD *(Algorithm 4 or Algorithm 5) with stepsize $\gamma \leq \frac{1}{2RL}$ satisfy the following bound:*

$$\frac{1}{K+1}\sum_{k=0}^{K}\mathbb{E}\left[\left\|x^k - x^{k-\delta^k}\right\|^2\right] \leq \frac{1}{2L^2(K+1)}\sum_{k=0}^{K}\mathbb{E}\left[\left\|\nabla f\left(x^{k-\delta^k}\right)\right\|^2\right] + \frac{\gamma}{L}\sigma^2.$$

*Proof.* Assume that we get a stochastic gradient from the worker with index $i_k$ when calculating $x^{k+1}$. We begin by

expanding the difference and applying the tower property, Assumption 1.3, Young's inequality, and Jensen's inequality:

$$
\mathbb{E}\left[\left\|x^k - x^{k-\delta^k}\right\|^2\right] = \mathbb{E}\left[\left\|\sum_{j=k-\delta^k}^{k-1} \gamma \nabla f\left(x^{j-\delta_j}; \xi_{i_j}^{j-\delta_j}\right)\right\|^2\right]
$$

$$
\leq 2\mathbb{E}\left[\left\|\sum_{j=k-\delta^k}^{k-1} \gamma \nabla f\left(x^{j-\delta_j}\right)\right\|^2\right] + 2\mathbb{E}\left[\left\|\sum_{j=k-\delta^k}^{k-1} \gamma \left(\nabla f\left(x^{j-\delta_j}; \xi_{i_j}^{j-\delta_j}\right) - \nabla f\left(x^{j-\delta_j}\right)\right)\right\|^2\right]
$$

$$
\leq 2\mathbb{E}\left[\left\|\gamma \sum_{j=k-\delta^k}^{k-1} \nabla f\left(x^{j-\delta_j}\right)\right\|^2\right] + 2\delta^k \gamma^2 \sigma^2
$$

$$
\leq 2\delta^k \gamma^2 \sum_{j=k-\delta^k}^{k-1} \mathbb{E}\left[\left\|\nabla f\left(x^{j-\delta_j}\right)\right\|^2\right] + 2\delta^k \gamma^2 \sigma^2.
$$

Using that $\gamma \leq \frac{1}{2RL}$ and $\delta^k \leq R - 1$ (by the design), we obtain:

$$
\mathbb{E}\left[\left\|x^k - x^{k-\delta^k}\right\|^2\right] \leq \frac{1}{2L^2 R} \sum_{j=k-\delta^k}^{k-1} \mathbb{E}\left[\left\|\nabla f\left(x^{j-\delta_j}\right)\right\|^2\right] + \frac{\gamma}{L}\sigma^2.
$$

Next, summing over all iterations $k = 0, \ldots, K$, we get:

$$
\sum_{k=0}^{K} \mathbb{E}\left[\left\|x^k - x^{k-\delta^k}\right\|^2\right] \leq \frac{1}{2L^2 R} \sum_{k=0}^{K} \sum_{j=k-\delta^k}^{k-1} \mathbb{E}\left[\left\|\nabla f\left(x^{j-\delta_j}\right)\right\|^2\right] + (K+1)\frac{\gamma}{L}\sigma^2.
$$

Observe that each squared norm $\left\|\nabla f\left(x^{j-\delta_j}\right)\right\|^2$ in the right-hand sums appears at most $R$ times due to the algorithms' design. Specifically, $\delta^k \leq R - 1$ for all $k \geq 0$, ensuring no more than $R$ squared norms appear in the sums. Therefore:

$$
\sum_{k=0}^{K} \mathbb{E}\left[\left\|x^k - x^{k-\delta^k}\right\|^2\right] \leq \frac{1}{2L^2} \sum_{k=0}^{K} \mathbb{E}\left[\left\|\nabla f\left(x^{k-\delta^k}\right)\right\|^2\right] + (K+1)\frac{\gamma}{L}\sigma^2.
$$

Finally, dividing the inequality by $K+1$ completes the proof. $\qquad\square$

## D. Proof of Theorem 5.1

**Theorem 5.1.** *Let Assumptions 1.1, 1.2, and 1.3 hold. Let the stepsize in* Ringmaster ASGD *(Algorithm 4 or Algorithm 5) be* $\gamma = \min\left\{\frac{1}{2RL}, \frac{\varepsilon}{4L\sigma^2}\right\}$, *and delay threshold* $R = \max\left\{1, \left\lceil\frac{\sigma^2}{\varepsilon}\right\rceil\right\}$. *Then, under the* universal computation model, Ringmaster ASGD *finds an* $\varepsilon$*–stationary point after at most* $T_{\bar{K}}$ *seconds, where* $\bar{K} := \left\lceil\frac{48L\Delta}{\epsilon}\right\rceil$ *and* $T_{\bar{K}}$ *is the* $\bar{K}^{th}$ *element of the following recursively defined sequence:*

$$
T_K := \min\left\{T \geq 0 : \sum_{i=1}^{n} \left\lfloor \frac{1}{4}\int_{T_{K-1}}^{T} v_i(\tau)d\tau \right\rfloor \geq R\right\}
$$

*for all* $K \geq 1$ *and* $T_0 = 0$.

*Proof.* From Theorem 4.1, the iteration complexity of Ringmaster ASGD is

$$
K = \left\lceil \frac{8RL\Delta}{\epsilon} + \frac{16\sigma^2 L\Delta}{\epsilon^2} \right\rceil. \tag{16}
$$

Without loss of generality, we assume that [5] $L\Delta > \varepsilon/2$. Thus, $\left\lceil \frac{8RL\Delta}{\epsilon} + \frac{16\sigma^2 L\Delta}{\epsilon^2} \right\rceil \leq \frac{16RL\Delta}{\epsilon} + \frac{32\sigma^2 L\Delta}{\epsilon^2}$ and

$$K \leq R \times \left\lceil \frac{K}{R} \right\rceil = R \times \left\lceil \frac{16L\Delta}{\epsilon} + \frac{32\sigma^2 L\Delta}{R\epsilon^2} \right\rceil.$$

Using the choice of $R$, we get

$$K \leq R \times \left\lceil \frac{48L\Delta}{\epsilon} \right\rceil.$$

In total, the algorithms will require $\left\lceil \frac{48L\Delta}{\epsilon} \right\rceil$ by $R$ consecutive updates of $x^k$ to find an $\varepsilon$–stationary point. Let us define $\bar{K} := \left\lceil \frac{48L\Delta}{\epsilon} \right\rceil$. Using Lemma 5.1, we know that Ringmaster ASGD requires at most

$$T_1 := T(R,0) = \min \left\{ T \geq 0 : \sum_{i=1}^{n} \left\lfloor \frac{1}{4} \int_0^T v_i(\tau)d\tau \right\rfloor \geq R \right\}$$

seconds to finish the first $R$ consecutive updates of the iterates. Since the algorithms will finish the *first* $R$ consecutive updates after at most $T_1$ seconds, they will start the iteration $R+1$ before time $T_1$. Thus, using Lemma 5.1 again, they will require at most

$$T_2 := T(R,T_1) = \min \left\{ T \geq 0 : \sum_{i=1}^{n} \left\lfloor \frac{1}{4} \int_{T_1}^T v_i(\tau)d\tau \right\rfloor \geq R \right\}$$

seconds to finish the first $2 \times R$ consecutive updates. Using the same reasoning, they will finish the first $\left\lceil \frac{48L\Delta}{\epsilon} \right\rceil \times R$ consecutive updates after at most

$$T_{\bar{K}} := T(R,T_{\bar{K}-1}) = \min \left\{ T \geq 0 : \sum_{i=1}^{n} \left\lfloor \frac{1}{4} \int_{T_{\bar{K}-1}}^T v_i(\tau)d\tau \right\rfloor \geq R \right\}$$

seconds. □

## E. Derivations for the Example from Section 2

Let $\tau_i = \sqrt{i}$ for all $i \in [n]$, then

$$T_{\mathrm{R}} = \Theta \left( \min_{m \in [n]} \left( \frac{1}{m} \sum_{i=1}^{m} \frac{1}{\sqrt{i}} \right)^{-1} \left( \frac{L\Delta}{\varepsilon} + \frac{\sigma^2 L\Delta}{m\varepsilon^2} \right) \right).$$

Using

$$\sum_{i=1}^{m} \frac{1}{\sqrt{i}} = \Theta \left( \sqrt{m} \right)$$

for $m \geq 1$, we simplify the term:

$$\left( \frac{1}{m} \sum_{i=1}^{m} \frac{1}{\sqrt{i}} \right)^{-1} = \Theta(\sqrt{m}),$$

$$T_{\mathrm{R}} = \Theta \left( \min_{m \in [n]} \sqrt{m} \left( \frac{L\Delta}{\varepsilon} + \frac{\sigma^2 L\Delta}{m\varepsilon^2} \right) \right) = \Theta \left( \min_{m \in [n]} \left( \frac{L\Delta\sqrt{m}}{\varepsilon} + \frac{\sigma^2 L\Delta}{\sqrt{m}\varepsilon^2} \right) \right).$$

The minimum is achieved when the two terms are balanced, i.e., at

$$m = \min \left\{ \left\lceil \frac{\sigma^2}{\varepsilon} \right\rceil, n \right\}.$$

---

[5]Otherwise, using $L$–smoothness, $\left\| \nabla f(x^0) \right\|^2 \leq 2L\Delta \leq \varepsilon$, and the initial point is an $\varepsilon$–stationary point

Substituting this value of $m$, we obtain:

$$T_{\mathrm{R}} = \Theta\left(\max\left[\frac{\sigma L\Delta}{\varepsilon^{3/2}}, \frac{\sigma^2 L\Delta}{\sqrt{n}\varepsilon^2}\right]\right).$$

We now consider $T_{\mathrm{A}}$:

$$T_{\mathrm{A}} = \Theta\left(\left(\frac{1}{n}\sum_{i=1}^{n}\frac{1}{\tau_i}\right)^{-1}\left(\frac{L\Delta}{\varepsilon} + \frac{\sigma^2 L\Delta}{n\varepsilon^2}\right)\right).$$

Using

$$\sum_{i=1}^{n}\frac{1}{\sqrt{i}} = \Theta\left(\sqrt{n}\right)$$

for $n \geq 1$, we simplify the term:

$$\left(\frac{1}{n}\sum_{i=1}^{n}\frac{1}{\sqrt{i}}\right)^{-1} = \Theta\left(\sqrt{n}\right).$$

Substituting this result into $T_{\mathrm{A}}$, we have:

$$T_{\mathrm{A}} = \Theta\left(\sqrt{n}\left(\frac{L\Delta}{\varepsilon} + \frac{\sigma^2 L\Delta}{n\varepsilon^2}\right)\right) = \Theta\left(\frac{L\Delta\sqrt{n}}{\varepsilon} + \frac{\sigma^2 L\Delta}{\sqrt{n}\varepsilon^2}\right) = \Theta\left(\max\left[\frac{L\Delta\sqrt{n}}{\varepsilon}, \frac{\sigma^2 L\Delta}{\sqrt{n}\varepsilon^2}\right]\right).$$

## F. When the Initial Point is an $\varepsilon$–Stationary Point

Under the assumption of $L$–smoothness (Assumption 1.1), we have:

$$f(y) \leq f(x) + \langle\nabla f(x), y - x\rangle + \frac{L}{2}\|y - x\|^2$$

for all $x, y \in \mathbb{R}^d$. Taking $y = x - \frac{1}{L}\nabla f(x)$,

$$f\left(x - \frac{1}{L}\nabla f(x)\right) \leq f(x) - \frac{1}{2L}\|\nabla f(x)\|^2.$$

Since $f\left(x - \frac{1}{L}\nabla f(x)\right) \geq f^{\mathrm{inf}}$ and taking $x = x^0$, we get

$$\left\|\nabla f(x^0)\right\|^2 \leq 2L\Delta.$$

Thus, if $2L\Delta \leq \varepsilon$, then $\left\|\nabla f(x^0)\right\|^2 \leq \varepsilon$.

## G. Experiments

Asynchronous SGD has consistently demonstrated its effectiveness and practicality, achieving strong performance in various applications (Recht et al., 2011; Lian et al., 2018; Mishchenko et al., 2022), along with numerous other studies supporting its utility. Our main goal of this paper was to refine the method further and establish that it is not only practical but also *theoretically optimal*.

At the same time, Tyurin & Richtárik (2023) found out that the previous version of Asynchronous SGD has slow convergence when the number of workers is large, and their computation performances are heterogeneous using numerical experiments (see Figure 1).

We now reproduce this experiment with the new Ringmaster ASGD method and compare it with the previous Asynchronous SGD (we call it Delay-Adaptive ASGD in our experiments and take the version from (Mishchenko et al., 2022)) and Rennala SGD methods. The optimization task is based on the convex quadratic function $f : \mathbb{R}^d \to \mathbb{R}$ such that

$$f(x) = \frac{1}{2}x^\top \mathbf{A}x - b^\top x \qquad \forall x \in \mathbb{R}^d,$$

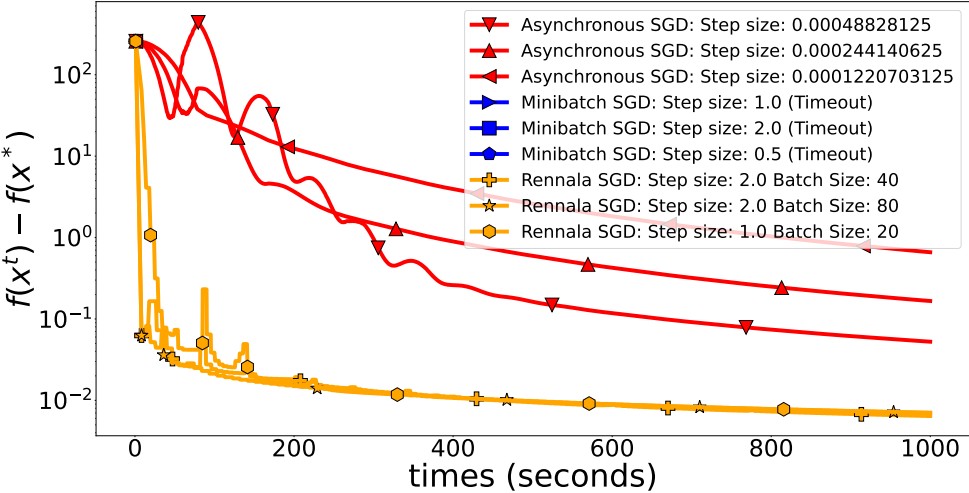

Figure 1: Experiment with $n = 10000$ from (Tyurin & Richtárik, 2023) showing the slow convergence of the previous Asynchronous SGD method.

where

$$
\mathbf{A} = \frac{1}{4} \begin{bmatrix} 2 & -1 & & 0 \\ -1 & \ddots & \ddots & \\ & \ddots & \ddots & -1 \\ 0 & & -1 & 2 \end{bmatrix} \in \mathbb{R}^{d \times d}, \quad b = \frac{1}{4} \begin{bmatrix} -1 \\ 0 \\ \vdots \\ 0 \end{bmatrix} \in \mathbb{R}^d.
$$

We set $d = 1729$ and $n = 6174$. Assume that all $n$ workers have access to the following unbiased stochastic gradients:

$$
\nabla f(x, \xi) = \nabla f(x) + \xi,
$$

where $\xi \sim \mathcal{N}(0, 0.01^2)$.

The experiments were implemented in Python. The distributed environment was emulated on machines with Intel(R) Xeon(R) Gold 6248 CPU @ 2.50GHz. The computation times for each worker are simulated as $\tau_i = i + |\eta_i|$ for all $i \in [n]$, where $\eta_i \sim \mathcal{N}(0, i)$. We tuned the stepsize from the set $\{5^p : p \in [-5, 5]\}$. Both the batch size for Rennala SGD and the delay threshold for Ringmaster ASGD were tuned from the set $\{\lceil n/4^p \rceil : p \in \mathbb{N}_0\}$. The experimental results are shown in Figure 2.

The obtained result confirms that Ringmaster ASGD is indeed faster than Delay-Adaptive ASGD and Rennala SGD in the considered setting. One can see the numerical experiments support that our theoretical results, and we significantly improve the convergence rate of the previous version of Asynchronous SGD (Delay-Adaptive ASGD).

### G.1. Neural Network Experiment

To show that our method also works well for neural networks, we trained a small 20-layer neural network with ReLU activation on the MNIST dataset (LeCun et al., 1998). We used the same number of workers as in the previous experiment ($n = 6174$) and kept the same time distributions. The results are shown in Figure 3.

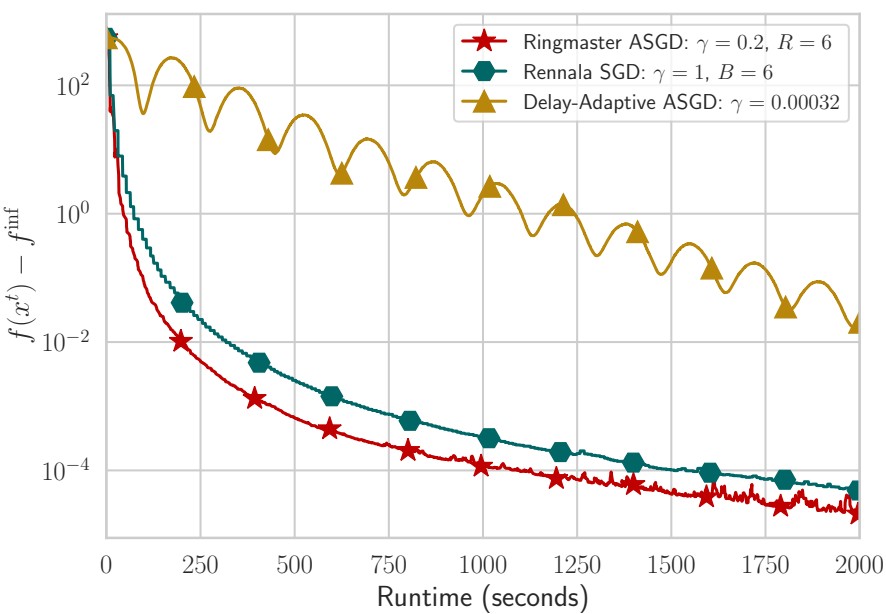

Figure 2: Experiment with $n = 6174$ and $d = 1729$ showing the convergence of Ringmaster ASGD, Delay-Adaptive ASGD, and Rennala SGD.

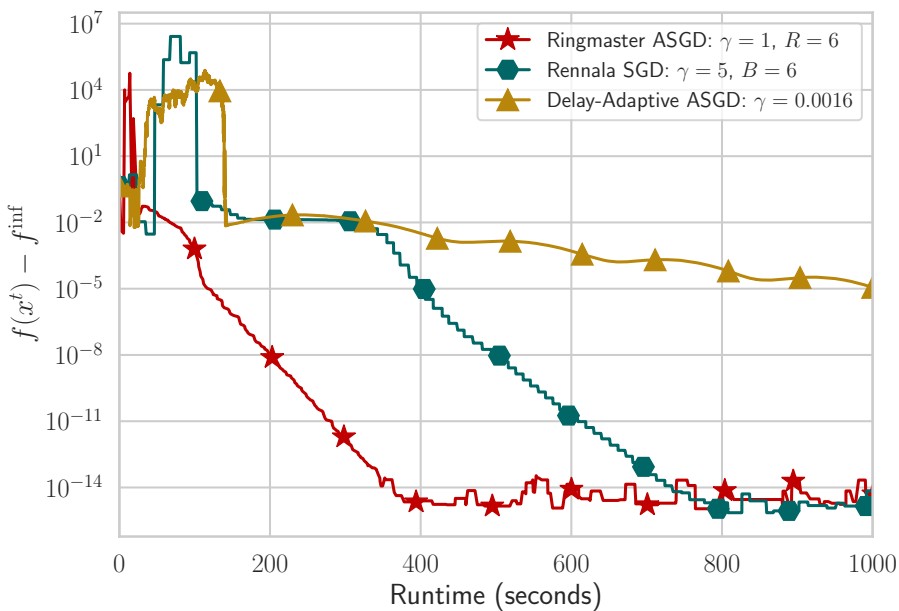

Figure 3: We run an experiment on a small 2-layer neural network with ReLU activation on the MNIST dataset, showing that our method, Ringmaster ASGD, is more robust and outperforms Delay-Adaptive ASGD and Rennala SGD.

