# OpenReview forum: "Ringmaster ASGD: The First Asynchronous SGD with Optimal Time Complexity"
_ICML.cc/2025/Conference — ICML 2025 poster_

### Official Review · Reviewer_HLwu · 2025-03-10

**Overall Recommendation:** 3

**Summary:**

This paper studied federated learning where each client has a different computation resource.
The authors first showed that it is optimal to run Asynchronous SGD on the fastest $m^\star$ clients. (Theorem 2.1)
This naive approach is optimal, but it does not work well when the computation power of each client changes over time.
Then, the authors proposed Ringmaster ASGD, showing that Ringmaster ASGD can also achieve the optimal convergence rate while it does not require prior knowledge of the computation power of each client.

**Claims And Evidence:**

The proposed method and the derived convergence results sound reasonable for the reviewer.

**Essential References Not Discussed:**

* The reviewer feels that the relationship between this paper and existing papers is a bit unclear. Some of the results shown in this paper have been already proposed in the existing papers. Specifically, Theorem 2.1 has been already shown in [1], while the reviewer feels that the authors claimed that these results are also novel results in this paper.

## Reference
[1] Alexander Tyurin, Peter Richtárik, Optimal Time Complexities of Parallel Stochastic Optimization Methods Under a Fixed Computation Model, NeurIPS 2023

**Experimental Designs Or Analyses:**

No experimental results are shown in this paper.

**Methods And Evaluation Criteria:**

Yes.

**Other Comments Or Suggestions:**

N/A

**Other Strengths And Weaknesses:**

See other sections.

**Questions For Authors:**

* In Sec. 2.2, the authors claimed that "Naively selecting the fastest $m^\star$ workers at the start of the method and keeping this selection unchanged may therefore lead to significant issues in practice". However, since there is no need to select the same clients over time, the reviewer is wondering if Algorithm 3 really does not work well when the client speeds change over time.
* The reviewer feels that this paper is claiming Algorithm 3 and Threom 2.1 are novel, while these results have already been shown in [1]. Thus, the main contribution of this paper is Sec. 3. The reviewer would like to suggest that the authors clarify which parts of this paper are novel.
* The primary contribution of this paper is theoretical, while the reviewer would like to suggest that the authors verify their results by demonstrating the experimental results.


## Reference
[1] Alexander Tyurin, Peter Richtárik, Optimal Time Complexities of Parallel Stochastic Optimization Methods Under a Fixed Computation Model, NeurIPS 2023

**Relation To Broader Scientific Literature:**

The proposed method, Ringmaster ASGD sounds novel, and it sounds novel that Ringmaster ASGD does not require the prior knowledge of computation powers of each client.

**Theoretical Claims:**

The reviewer did not check the proof, while the claim of this paper sounds reasonable.

---

> ### Author Rebuttal · Authors · 2025-03-29
>
> We thank the reviewer for the review.
>
> > This paper studied federated learning where each client has a different computation resource.
>
> Our setup is relevant not only to Federated Learning but also to datacenter environments, where heterogeneous GPU clusters are common. Even in datacenters with identical GPUs, failures become inevitable as the number of GPUs increases, introducing additional heterogeneity. For example, see [0], Section 3.3.4.
>
> [0] Grattafiori, Aaron, et al. "The llama 3 herd of models." arXiv preprint arXiv:2407.21783 (2024).
>
>
> > No experimental results are shown in this paper.
>
> Experiments are presented in Section F of the appendix. We included them there because the primary focus of our paper was to refine the method and establish that it is not only practical but also theoretically optimal. That said, Figures 1 and 2 further demonstrate the practicality of our method, with Figure 2 specifically showing that it outperforms existing benchmarks in speed.
>
>
> > The reviewer feels that the relationship between this paper and existing papers is a bit unclear. Some of the results shown in this paper have been already proposed in the existing papers. Specifically, Theorem 2.1 has been already shown in [1], while the reviewer feels that the authors claimed that these results are also novel results in this paper.
>
> Let us clarify the relationship between [1] and our work. [1] establishes lower bounds for the time complexity of first-order methods, while we propose a fully asynchronous method and derive an upper bound that matches these lower bounds. Additionally, [1] analyzes Rennala SGD, which also attains the lower bound, but the key difference is that Rennala SGD is not fully asynchronous—it operates as Minibatch SGD (which performs synchronous model updates) combined with an asynchronous minibatch collection mechanism. We discuss this distinction in detail in Section 1.2. Our main contribution lies in closing the gap by developing a fully asynchronous method that achieves the lower bound on time complexity.
>
> Regarding Theorem 2.1, it is not explicitly stated in prior work, though it follows as a direct consequence of existing results. The analysis of Asynchronous SGD (Algorithm 1) appears in [2] and [3], while [1] establishes its time complexity. We formally state this in lines 175-183 and then show that Theorem 2.1 naturally follows with a different choice of workers, leading to the optimal time complexity. Notably, this specific choice of workers and its role in achieving optimal time complexity has not been shown in prior work.
>
> [1] Alexander Tyurin, Peter Richtárik, Optimal Time Complexities of Parallel Stochastic Optimization Methods Under a Fixed Computation Model, NeurIPS 2023
>
> [2] Koloskova, A., et al. Sharper convergence guarantees for Asynchronous SGD for distributed and federated learning. NeurIPS 2022.
>
> [3] Mishchenko, K., et al. Asynchronous SGD beats minibatch SGD under arbitrary delays. NeurIPS 2022.
>
>
> > In Sec. 2.2, the authors claimed that "Naively selecting the fastest workers at the start of the method and keeping this selection unchanged may therefore lead to significant issues in practice". However, since there is no need to select the same clients over time, the reviewer is wondering if Algorithm 3 really does not work well when the client speeds change over time.
>
> It is possible to reselect workers each time the $\tau_i$ values change, but this would introduce additional computational overhead due to the need for continuous selection. Plus, it is not obvious when we should reselect workers. Additionally, a bigger issue is that $\tau_i$ is typically unknown, making this approach impractical. Instead, we propose a simpler solution using a threshold-based approach, as presented in Algorithms 4 and 5, which does not require $\tau_i$ and automatically and adaptively chooses the fastest subsets of workers.
>
>
> > The reviewer feels that this paper is claiming Algorithm 3 and Threom 2.1 are novel, while these results have already been shown in [1]. Thus, the main contribution of this paper is Sec. 3. The reviewer would like to suggest that the authors clarify which parts of this paper are novel.
>
> No, these results are not shown in prior work, as discussed above.
>
>
> > The primary contribution of this paper is theoretical, while the reviewer would like to suggest that the authors verify their results by demonstrating the experimental results.
>
> Indeed, our work is primarily theoretical, as mentioned earlier. However, we provide experimental results in Section F of the appendix.

---

### Official Review · Reviewer_kbLE · 2025-03-11

**Overall Recommendation:** 2

**Summary:**

This paper discusses the characteristics of asynchronous parallel algorithms when a delay upper bound is provided, specifically Algorithm 4 and Algorithm 5. Asynchronous parallel SGD was the focus of SGD research from 2014 to 2020. With the popularity of the Adam algorithm, research on SGD has begun to decline. This paper offers rigorous proofs for Algorithm 4 and Algorithm 5, and simple experiments are provided in the appendix.

I believe the biggest issue with this paper is that it merely adds a delay constraint on top of Algorithm 3 without altering any of its other mathematical properties. Providing such an analysis requires only minor modifications to the analysis of Algorithm 3. Therefore, the paper should have extensively discussed the shortcomings of Algorithm 3, particularly in Section 2.2. However, the content in Section 2.2 does not convince me that Algorithm 3 has significant flaws. Furthermore, the analysis of Algorithm 3 itself, such as that of the Hogwild! algorithm, already includes discussions on maximum delay. Consequently, I find the contribution of this paper to be insufficient.

**Claims And Evidence:**

This paper does indeed present two asynchronous parallel algorithms and provides relatively complete proofs and very simple experiments for these algorithms. However, from an experimental perspective, it is difficult to support the arguments made in this paper.

**Essential References Not Discussed:**

no

**Experimental Designs Or Analyses:**

There is almost no experiments in this paper

**Methods And Evaluation Criteria:**

From a theoretical perspective, this paper provides a fairly complete proof. From an experimental perspective, it is difficult to support the arguments made in this paper.

**Other Comments Or Suggestions:**

no

**Other Strengths And Weaknesses:**

This paper discusses an asynchronous parallel algorithm with a bounded delay, which essentially adds a delay constraint to Algorithm 3. The proof and analysis of Algorithm 3 are already very well-established, and in these proof,  the requirement for delay in these proofs is based on maximum delay. Therefore, the algorithm and analysis presented in this paper have limited value.

In practical industrial scenarios, significant fluctuations in node delays are relatively rare. Spending additional computational resources to control delay may result in higher performance losses than those caused by the delay itself. Hence, from both algorithm design and theoretical analysis perspectives, the value of this paper is quite limited.

**Questions For Authors:**

Show more  discussion about  the analysis difference between tradtional ASGD and algorithm 4/5.

**Relation To Broader Scientific Literature:**

The main conclusions of this paper are closely related to the analysis of Algorithm 3. Based on the analysis of Algorithm 3, such as the Hogwild! algorithm analysis, I believe that the analysis presented in this paper is quite straightforward.

**Theoretical Claims:**

I I have basically reviewed the proof frame approach.

---

> ### Author Rebuttal · Authors · 2025-03-29
>
> Thank you for the review.
>
> > Asynchronous parallel SGD was the focus of SGD research from 2014 to 2020. With the popularity of the Adam algorithm, research on SGD has begun to decline.
>
> We respectfully disagree with this statement. One of the important works [1] in stochastic optimization (a field that includes both SGD and Adam) was published only in 2023 (preprint in 2019). [1] proves that SGD is an optimal method in the nonconvex stochastic setting, meaning that, theoretically, Adam cannot be better than SGD. At the same time, we agree that Adam is a very important method that requires further investigation, but it is based on SGD. Without understanding SGD, it would be much more difficult to understand Adam.
>
> > I believe the biggest issue with this paper is that it merely adds a delay constraint on top of Algorithm 3 without altering any of its other mathematical properties.
>
> Adding this delay constraint in Alg. 4 is a non-trivial and elegant fix for the original Async SGD. Surprisingly, this simple yet essential algorithmic improvement was previously overlooked.
>
> > Furthermore, the analysis of Algorithm 3 itself, such as that of the Hogwild! algorithm, already includes discussions on maximum delay  .. The main conclusions of this paper are closely related to the analysis of Algorithm 3. Based on the analysis of Algorithm 3, such as the Hogwild! algorithm analysis, I believe that the analysis presented in this paper is quite straightforward ...
>
> Note that the analysis of Async SGD in the Hogwild! paper is not the tightest previous result. [2,3] establish a better rate. We further improve it to the optimal time complexity, which is unimprovable due to the lower bounds.
>
> The mathematical properties change significantly. Instead of the classical asynchronous SGD, we analyze a new version in which outdated and irrelevant data are ignored. First, we analyze the sum $\sum_{k=0}^K \mathbb{E}[|x^k - x^{k-\delta^k} |^2]$ in Lemma C.2 more carefully and improve upon the analysis in [2,3]. Next, Lemma 4.1, Theorem 4.2, Lemma 5.1, and Theorem 5.1 are entirely new results, which show the optimality of the Async SGD approach for the first time in the literature!
>
> > Providing such an analysis requires only minor modifications to the analysis of Algorithm 3. Therefore, the paper should have extensively discussed the shortcomings of Algorithm 3 ... the content in Section 2.2 does not convince me that Algorithm 3 has significant flaws.
>
> Algorithm 3 presents at least two key flaws. First, it requires computation times as input and assumes that these times are fixed, an assumption that is clearly unrealistic in practical scenarios. Second, as detailed in Section 2.2, there is another significant flaw: Algorithm 3 is neither robust nor adaptive to adversarial computation environments.
>
> > This paper does indeed present two asynchronous parallel algorithms and provides relatively complete proofs and very simple experiments for these algorithms. However, from an experimental perspective, it is difficult to support the arguments made in this paper.
>
> The goal of this paper was to prove the theoretical optimality of the Asynchronous SGD approach. This paper closes an important theoretical question from the optimization field. This paper should be read primarily as a theoretical paper.
>
> > In practical industrial scenarios, significant fluctuations in node delays are relatively rare.
>
> This observation does not always hold; significant fluctuations appear in federated learning. Even when fluctuations are minimal, we still obtain improved and optimal theoretical guarantees (see Table, Thm. 4.2, 5.1).
>
> > Spending additional computational resources to control delay may result in higher performance losses than those caused by the delay itself.
>
> Additional computations due to control of the delays are negligible compared to stochastic gradient computation times. How can a simple comparison $\delta^k < R$ of two integers decrease the performance of modern computation systems?
>
> > Hence, from both algorithm design and theoretical analysis perspectives, the value of this paper is quite limited.
>
> **We strongly believe obtaining an optimal Async SGD method is an important task for the ICML community. This is the first paper to demonstrate the optimality of asynchronous SGD in terms of time complexity—a point we believe was overlooked by the reviewer. The corresponding complexity is significantly better than that of previous variants (see Table 1). We believe this work closes an important open problem in asynchronous optimization.**
>
> [1] Arjevani, Y, et al. "Lower bounds for non-convex stochastic optimization." Mathematical Programming 199.1 (2023): 165-214.
>
> [2] Koloskova, A., et al. Sharper convergence guarantees for Asynchronous SGD for distributed
> and federated learning. NeurIPS 2022.
>
> [3] Mishchenko, K., et al. Asynchronous SGD beats minibatch SGD under arbitrary delays. NeurIPS 2022.

---

> > ### Comment · Reviewer_kbLE · 2025-04-07
> >
> > The author's rebuttal and other reviewers' opinions addressed some of my concerns. I do think that this approach might have some practical significance in the field of federated learning, so I have slightly increased my score. However, the improvements to asynchronous SGD mentioned in the paper are very minimal, if not expected. Table 1 shows that the modifications and optimizations are basically built upon the work of Koloskova et al., thus, I believe the innovation is quite limited. From the perspective of clusters and data centers, this approach is rather conventional. As I mentioned in my review, the impact of delays has long been detailed in theorems in early studies of asynchronous SGD, which already guides the choice of cluster size (since the minimum delay equates to maximum the number of workers, i.e.,  parallelism). For instance, in industrial applications like AD  CTR model training, cluster sizes typically do not exceed 300 workers because increased parallelism leading to higher delays would severely hinder convergence speed.
> >
> > Regarding experiments, theoretically, I expect to see a match between theoretical curves and experimental results (since the experiments in this paper are based on very simple analyzable functions). Practically, I hope to see outcomes from neural networks, even relatively simple ones like ResNet20. Considering the main contribution of this paper lies in its theory, I am merely offering suggestions; issues with experiments are not the reason for my disapproval of this paper.

---

> > > ### Author Response · Authors · 2025-04-07
> > >
> > > Thank you for increasing the score! Let us clarify the remaining concerns:
> > >
> > > > However, the improvements to asynchronous SGD mentioned in the paper are very minimal, if not expected. Table 1 shows that the modifications and optimizations are basically built upon the work of Koloskova et al., thus, I believe the innovation is quite limited.
> > >
> > > It is true that our work builds upon the results of Koloskova et al., but this is a natural progression in mathematical and optimization sciences. There is a connection between the work of Koloskova et al. and the earlier work [1], which in turn may rely on techniques introduced in [2], and so on. All these papers, including ours, study the same method, each improving upon the results of the previous ones. Our work builds on Koloskova et al., their work may build on [1], [1] may build on [2], and so forth.
> > >
> > > In this line of progress, we provide a new analysis that improves all previous results. Furthermore, we show that our analysis is tight and can not be improved further by proving matching lower bounds, which we believe is an important contribution.
> > >
> > > ---
> > >
> > > > As I mentioned in my review, the impact of delays has long been detailed in theorems in early studies of asynchronous SGD, which already guides the choice of cluster size (since the minimum delay equates to maximum the number of workers, i.e., parallelism). For instance, in industrial applications like AD CTR model training, cluster sizes typically do not exceed 300 workers because increased parallelism leading to higher delays would severely hinder convergence speed.
> > >
> > > Note that in modern large-scale training, the number of workers can far exceed 300. For example, Llama 3 was reportedly trained using 16,000 GPUs. At such a scale, delays and interruptions become increasingly common. Moreover, the number of GPUs used in training continues to grow and is rapidly approaching 100,000 (if it hasn’t already).
> > >
> > > In [3], a 16K GPU cluster experienced 419 unexpected interruptions over a 54-day period. The following excerpt from the paper illustrates this: "During a 54-day snapshot period of pre-training, we experienced a total of 466 job interruptions. Of these, 47 were planned interruptions due to automated maintenance operations such as firmware upgrades or operatorinitiated operations like configuration or dataset updates. The remaining 419 were unexpected interruptions, which are classified in Table 5."
> > >
> > > For more details, see [3], Section 3.3.4.
> > >
> > > As the number of GPUs increases, delays and interruptions become more frequent; our work offers both new practical and theoretical guidance on how to effectively manage significant delays.
> > >
> > > ---
> > >
> > > We believe that establishing optimality and improving the results of [1, 2, 4] (and many other related works) for Asynchronous SGD is an important objective for the ICML community. Our work is the first to achieve optimal time complexity — a key contribution that may have been overlooked by the reviewer.
> > >
> > > ---
> > >
> > > > Regarding experiments, theoretically, I expect to see a match between theoretical curves and experimental results (since the experiments in this paper are based on very simple analyzable functions). Practically, I hope to see outcomes from neural networks, even relatively simple ones like ResNet20. Considering the main contribution of this paper lies in its theory, I am merely offering suggestions; issues with experiments are not the reason for my disapproval of this paper.
> > >
> > > We ran a small MNIST experiment using a 2-layer neural network with ReLU activation. See results
> > > [here](https://anonymous.4open.science/api/repo/nn_exp-17E3/file/real_data.pdf?v=695e36eb)
> > >
> > > ---
> > >
> > > Thank you for your review!
> > >
> > > Best regards,
> > > Authors
> > >
> > > ---
> > >
> > >
> > > [1] Sebastian U. Stich and Sai Praneeth Karimireddy. The error-feedback framework: SGD with delayed
> > > gradients. Journal of Machine Learning Research, 21(237):1–36, 2020
> > >
> > > [2] Alekh Agarwal and John C Duchi. Distributed delayed stochastic optimization. In Advances in Neural
> > > Information Processing Systems 24, pages 873–881. Curran Associates, Inc., 2011.
> > >
> > > [3] Grattafiori, Aaron, et al. “The llama 3 herd of models.” arXiv preprint arXiv:2407.21783 (2024).
> > >
> > > [4] Koloskova, A., et al. Sharper convergence guarantees for Asynchronous SGD for distributed and federated learning. NeurIPS 2022.

---

### Official Review · Reviewer_kWtQ · 2025-03-14

**Overall Recommendation:** 3

**Summary:**

This paper proposes a method called Ringmaster ASGD to achieve the optimal time complexity for asynchronous methods as described in [1]. Ringmaster ASGD is a simple modification of vanilla asynchronous SGD. In Ringmaster ASGD, gradients with large delays (>R) are discarded.

[1] Tyurin, A. and Richt´arik, P. Optimal time complexities of parallel stochastic optimization methods under a fixed computation model. Advances in Neural Information Processing Systems,36,2023.

**Claims And Evidence:**

The claims are partially supported by evidence. Please see details below.

**Essential References Not Discussed:**

No.

**Experimental Designs Or Analyses:**

Only one very simple problem, a simulated convex problem, is used in experiment. Furthermore, only one specific value of R is evaluated. More experiments with more complex models and real datasets, under different settings of R, are needed to validate the efficiency of Ringmaster ASGD.

**Methods And Evaluation Criteria:**

The proposed methods and evaluation criteria make sense for the problem.

**Other Comments Or Suggestions:**

No.

**Other Strengths And Weaknesses:**

Algorithm 5 is a modification of Algorithm 4 that stops irrelevant computations. The update rules of these two algorithms are equivalent. Hence, the statement from line 319 to line 328 is sufficient, and the details of Algorithm 5 can be moved to the appendix for simplicity.

**Questions For Authors:**

Please refer to the above issues.

**Relation To Broader Scientific Literature:**

The key contributions of the paper are related to federated learning.

**Theoretical Claims:**

As shown in Theorem 4.2 and Theorem 5.1, the value of the delay threshold does not depend on the computation times. Does this imply that the same value of R can be used across different distributed systems? This conclusion is somewhat confusing and counter-intuitive. It seems that the optimal value of R should be related to the computing capability of the workers in the cluster.

---

> ### Author Rebuttal · Authors · 2025-03-31
>
> Thank you for the review.
>
> > As shown in Theorem 4.2 and Theorem 5.1, the value of the delay threshold does not depend on the computation times. Does this imply that the same value of R can be used across different distributed systems? This conclusion is somewhat confusing and counter-intuitive. It seems that the optimal value of R should be related to the computing capability of the workers in the cluster.
>
> To clarify, in the proof of Theorem 4.2, we select $R$ to minimize the upper bound on time complexity (Equation 11), up to a universal constant. The exact minimizer—without this constant—depends on the worker times $\tau_i$. However, the resulting time complexity differs only by a constant factor.
>
> Consider an optimization problem with varying worker times $\tau_i$. Let’s examine two extreme cases:
> - $\tau_i = \infty$ for all $i > 1$ and $\tau_1 = \tau > 0$ (only one active worker) – The optimal choice here is simply $R = 1$.
> - All $\tau_i$ are equal – The optimal $R$ in this case is potentially larger than $1$.
>
> While the optimal $R$ varies across these scenarios, the final time complexities remain within a constant factor of each other.
>
> This property is, in fact, quite powerful: as you pointed out, the same value of $R$ can be used across different data centers, leading to, at most a constant-factor difference in performance. We appreciate this observation and will clarify it further in the camera-ready version of the paper.
>
> > Only one very simple problem, a simulated convex problem, is used in experiment. Furthermore, only one specific value of R is evaluated. More experiments with more complex models and real datasets, under different settings of R, are needed to validate the efficiency of Ringmaster ASGD.
>
> The primary goal of this paper is to establish the **theoretical optimality** of Asynchronous SGD. Our work addresses a fundamental open problem in optimization, providing the first proof of **optimal time complexity** for asynchronous SGD. We firmly believe that developing an optimal Asynchronous SGD method is essential for the ICML community. This work represents a significant step in that direction, offering theoretical guarantees that pave the way for future practical advancements.

---

> > ### Comment · Reviewer_kWtQ · 2025-04-08
> >
> > Thank the authors for the rebuttal.
> >
> > From the rebuttal, it is difficult to get the conclusion that the resulting time complexity differs only by a constant factor. Furthermore, it is counter-intuitive that the delay threshold does not depend on the computation times and the number of nodes in the cluster. Given a special case that the computation times are totally different for each node, it seems that a more reasonable choice is to set the delay threshold to be proportional to the number of nodes.
> >
> > In addition, even experiments on simple non-convex problems like neural networks with two or three layers, not necessarily large models on large datasets, can improve the convincingness of the paper. For the convex problem in the paper, other methods like variance-reduction based asynchronous SGD can achieve much faster convergence than the method proposed in this paper.

---

> > > ### Author Response · Authors · 2025-04-08
> > >
> > > Thank you for engaging with us. It seems there may be a misunderstanding, and we are happy to clarify.
> > >
> > > > From the rebuttal, it is difficult to get the conclusion that the resulting time complexity differs only by a constant factor.
> > >
> > > In Theorem 4.2, we provide an upper bound on the time complexity (eq. 8), expressed using big $\mathcal{O}$ notation, which hides universal constants (i.e., constants independent of any problem parameters). Reference [1] (specifically Theorem 6.4) gives a matching lower bound for the same class of functions (smooth, nonconvex) for first-order asynchronous methods. This lower bound coincides with our upper bound in eq. 8, up to universal constants.
> > >
> > > This implies that our method, with the threshold selection from Theorem 4.2, is unimprovable up to constant factors. In other words, while it is possible that for certain functions and specific time distributions ($\tau_i$), a different choice of the threshold $R$ might lead to better time complexity, the improvement can only be by a constant factor (e.g., twice as fast). It cannot exceed that because of the matching lower bound established in [1].
> > >
> > > For example, in the case where $\tau_i=\infty$ for $i >1$, the optimal choice of $R$ is 1, which gives a better time complexity than the choice in Theorem 4.2. However, even in this extreme case, the improvement remains within a constant factor.
> > >
> > > ---
> > > [1] Tyurin A., Richtárik P. Optimal Time Complexities of Parallel Stochastic Optimization Methods Under a Fixed Computation Model. NIPS 2023.
> > >
> > > > Furthermore, it is counter-intuitive that the delay threshold does not depend on the computation times and the number of nodes in the cluster.
> > >
> > > In fact, the optimal choice of $R$ does depend on the computation times, as we previously mentioned. However, there is no closed-form expression for the best $R$ in general—it depends on the specific values of $\tau_i$ and must be determined on a case-by-case basis.
> > >
> > > If desired, the optimal threshold $R$ can be written as
> > > $$
> > > \arg\min_{R\geq1}\left\\{t(R)\left(1+\frac{\sigma^2}{R\varepsilon}\right)\right\\},
> > > $$
> > > which comes from eq. 11 after removing constants that do not depend on $R$. Here, $t(R)$ denotes the time required for $R$ consecutive iterations, and it is upper bounded by eq. 7.
> > >
> > > Plugging in the bound from eq. 7, we obtain the following expression for the optimal $R$
> > > $$R = \max\left\\{\sigma\sqrt{\frac{m^*}{\varepsilon}},1\right\\},$$
> > > where
> > > $$m^*=\arg\min_{m\in [n]}\left\\{\left(\frac{1}{m}\sum_{i=1}^m\frac{1}{\tau_i}\right)^{-1}\left(1+2\sqrt{\frac{\sigma^2}{m\varepsilon}}+\frac{\sigma^2}{m\varepsilon}\right)=:T_m\right\\}.$$
> > >
> > > As you can see, the optimal $R$ depends on $m^*$, which in turn depends on the time distribution through the $\tau_i$ values.
> > >
> > > The choice of $R$ in Theorem 4.2 was made for simplicity—it has a closed-form expression and avoids dependence on the time distribution. While not always optimal, it achieves time complexity within a small constant factor of the best possible. This makes it a practical and robust choice, especially in the dynamic setting considered in Theorem 5.1.
> > >
> > > ---
> > > > Given a special case that the computation times are totally different for each node, it seems that a more reasonable choice is to set the delay threshold to be proportional to the number of nodes.
> > >
> > > Could you please clarify what you mean by ‘totally different’?
> > >
> > > In the rebuttal, we gave an example where $\tau_i=\infty$ for $i>1$, and the optimal choice is $R = 1$, which clearly does not depend on $n$. Even if the $\tau_i$ values are not infinity but are just very large (larger than the total convergence time using only one worker) and arbitrarily different from each other, the threshold is still $R=1$.
> > >
> > > Let’s consider a less extreme case: suppose $\tau_i=i^p$ for any $p\geq1$. Using the formula for $R$ based on $m^*$, we see that $T_m$ becomes an increasing function of $m$ beyond a certain $m$, so $m^*<n$. Consequently, even as the number of nodes increases, the value of $m^*$—and thus the optimal $R$—does not grow. Again, the delay threshold is not intrinsically tied to the total number of nodes.
> > >
> > > Finally, the idea that $R$ should scale with the number of workers is exactly what prior works assumed. For instance, [2] shows that the average delay grows with $n$ and proposes a delay-adaptive method to improve the convergence rate (eq. 4). However, our work argues that including updates from all workers may hinder convergence. We demonstrate that carefully controlling the delay threshold, even if it means excluding slower workers, leads to a faster rate (eq. 3). This highlights a key insight of our work: the delay threshold should not be simply scaled with the number of workers.
> > >
> > > > On experiments
> > >
> > > We did extra experiments with NN
> > > [here](https://anonymous.4open.science/api/repo/nn_exp-17E3/file/real_data.pdf?v=695e36eb).
> > >
> > > ---
> > > [2] Koloskova, A., et al. Sharper convergence guarantees for Asynchronous SGD for distributed and federated learning. NIPS 2022.

---

### Official Review · Reviewer_Mp5i · 2025-03-20

**Overall Recommendation:** 3

**Summary:**

In the settings when all clients compute the same function, the paper introduces a family of Asynchronous SGD algorithms:

-- A trivial algorithm which chooses the optimal number of fastest machines. The paper shows that this algorithm achieves the optimal convergence rate. The downside of the algorithm is that it doesn’t handle the case when clients have variable performance.

-- Two algorithms which ignore or cut off computations which last more than some specified number of rounds. For these algorithms, the paper shows optimal convergence rate for the settings when the client performances are constant and when they are variable.

**Claims And Evidence:**

Yes

**Essential References Not Discussed:**

Ok

**Experimental Designs Or Analyses:**

Ok

**Methods And Evaluation Criteria:**

Yes

**Other Comments Or Suggestions:**

I would like to recommend restructuring the paper as follows:

-- Move some of the discussion right before 1.1 to “Related work”

-- Move section 1.3 earlier, before section 1.1

-- Move paragraph “Why do we ignore the old stochastic gradients?” earlier, maybe around Equation (3)

-- You might want to clarify that there are other definitions of ε-stationary point (in particular, ones which would use ε^2)

-- You don’t refer to Table 1 in the text

-- Footnote 2 might be not obvious to a reader

-- You assume that all v_i are continuous. This might be an unrealistic assumption; moreover, I don’t think you use this assumption.

-- In Theorem 4.2, since you know the value of R, you can simplify the statement

**Other Strengths And Weaknesses:**

Ok

**Questions For Authors:**

Your algorithm uses a binary decision for every gradient: use it if the delay is less than R, and ignore it otherwise. This is fine if one knows a proper value of R, which introduces an additional hyperparameter. Do you think it’s possible to avoid using such a hyperparameter, e.g. by scaling the gradients inversely proportionally to the delay?

**Relation To Broader Scientific Literature:**

Ok

**Theoretical Claims:**

-- I don’t think paragraph “Proof techniques” actually provides any insights about the proof

-- The techniques feel incremental, with the main proof being completed mainly using Lemmas from Koloskova et al (2022).

---

> ### Author Rebuttal · Authors · 2025-03-31
>
> Thank you for the review.
>
> > The techniques feel incremental, with the main proof being completed mainly using Lemmas from Koloskova et al (2022).
>
> We acknowledge that the proof is not complicated, but we see this as an advantage rather than a limitation. A small yet impactful change can often be more valuable than a more complex modification achieving the same result.
>
> Our work provides the first proof of the optimality of asynchronous SGD in the literature. We establish this through a simple yet non-trivial and elegant idea: introducing a threshold on gradient delays. Surprisingly, this fundamental algorithmic improvement had been previously overlooked.
>
> Moreover, our approach significantly alters the mathematical properties of the method. Unlike classical asynchronous SGD, it discards outdated and irrelevant data, leading to a refined theoretical analysis. Specifically, we analyze the sum $\sum_{k=0}^K \mathbb{E}[|x^k - x^{k-\delta^k} |^2]$ in Lemma C.2 more carefully, improving upon the analysis in [1,2]. Additionally, Lemma 4.1, Theorem 4.2, Lemma 5.1, and Theorem 5.1 present entirely new results.
>
> > I would like to recommend restructuring the paper as follows:
>
> Thank you for the suggestions. We will make the changes for the camera-ready version of the paper.
>
> > You assume that all $v_i$ are continuous. This might be an unrealistic assumption; moreover, I don't think you use this assumption.
>
> We assume that $v_i$ is non-negative and continuous almost everywhere. It means that $v_i$ can be discontinued in a countable set of points, and our analysis still works. We believe that this is general enough since it allows the computation powers to "jump" on a countable set of times. We use this assumption (non-explicitly) when integrating $v_i$. Under this assumption, $v_i$ is Riemann integrable. Notice that we can easily assume that $v_i$ is measurable and use the Lebeague integral, but for clarity, we work with the Riemann integral.
>
> > Your algorithm uses a binary decision for every gradient: use it if the delay is less than R, and ignore it otherwise. This is fine if one knows a proper value of R, which introduces an additional hyperparameter. Do you think it's possible to avoid using such a hyperparameter, e.g. by scaling the gradients inversely proportionally to the delay?
>
> We think it may be possible to make $R$ adaptive by estimating it in an online fashion, but this is beyond the scope of our current research and is left for future work.
>
> Regarding scaling. Prior work [1,2] explored this approach by scaling gradients inversely proportionally to their delay, a method known as delay-adaptive ASGD. However, this does not achieve optimal time complexity—some outdated gradients must be ignored. We discuss this in the “Comparison to Previous Work” section (lines 306-318).
>
> That said, introducing an additional hyperparameter is unavoidable. However, the choice of $R$ is not particularly sensitive. Specifically, in Theorem 4.2, setting $R = \max\\{1, \lceil c \frac{\sigma^2}{\varepsilon} \rceil \\}$ for any absolute constant $c$ still ensures optimal time complexity up to a constant factor depending on $c$.
>
> [1] Koloskova, A., et al. Sharper convergence guarantees for Asynchronous SGD for distributed and federated learning. NeurIPS 2022.
>
> [2] Mishchenko, K., et al. Asynchronous SGD beats minibatch SGD under arbitrary delays. NeurIPS 2022.

---

### Decision · Program_Chairs · 2025-05-01

**Decision:**

Accept (poster)

**Comment:**

The paper gives a new complexity analysis of asynchronous SGD, improving upon the previous bounds given in Koloskova et al. The algorithm controls gradient staleness by discarding updates that are stale beyond a threshold R. The bounds provided by the paper match order-wise with lower bounds, thus ensuring that the analysis is tight. Here are some concerns raised by reviewers. The authors addressed these fairly well in the rebuttal prompting some reviewers to increase their scores.
* There are no experiments in the submitted version of the paper. During the rebuttal, the authors added some simple experiments, however, these could be further improved by considering larger neural networks and real-world datasets.
* Since the techniques are similar to Koloskova et al, there are some concerns about novelty.
* While the theoretical contribution of this work is solid, the practical value might be limited since most industry implementations are moving back to synchronous SGD. Asynchronous SGD could be of value in edge ML settings.